# BLUE: Bi-layer Heterogeneous Graph Fusion Network for Avian Influenza Forecasting

## Abstract

Accurate forecasting of avian influenza outbreaks within wild bird populations requires models that account for complex, multi-scale transmission patterns driven by various factors. Spatio-temporal GNN-based models have recently gained traction for infection forecasting due to their ability to capture relations and flow between spatial regions, but most existing frameworks rely solely on spatial regions and their connections. This overlooks valuable genetic information at the case level, such as cases in one region being genetically descended from strains in another, which is essential for understanding how infectious diseases spread through epidemiological linkages beyond geography. We systemically formulate AIV forecasting problem by proposing a **Bi-L**ayer heterogeneous graph f**U**sion pip**E**line (**BLUE**). This pipeline integrates genetic, spatial, and ecological data to achieve highly accurate outbreak forecasting. It 1) defines heterogeneous graphs from multiple information sources and multiple layers, 2) smooths across relation types, 3) performs fusion while retaining structural patterns, and 4) predicts future outbreaks via an autoregressive graph sequence model that captures transmission dynamics over time. To facilitate further research, we introduce **Avian-US** dataset, the dataset for avian influenza outbreak forecasting in the United States, incorporating genetic, spatial, and ecological data across locations. **BLUE** achieves superior performance over existing baselines, highlighting the value of incorporating multi-layer information into infectious disease forecasting. The code is available at: https://anonymous.4open.science/r/BLUE-60F8/README.md.

## 1 Introduction

Predicting the transmission of Avian Influenza Virus (AIV) remains a critical challenge in epidemiological research, due to the virus's capacity for widespread dissemination among avian populations. Increasing cross-species transmission poses serious risks to public health infrastructure and global biosecurity. Accurately predicting where outbreaks may occur is vital for initiating early interventions that reduce infection risk. To enable timely intervention and reduce the risk of zoonotic spillover, it is essential to identify high-risk regions prior to outbreak emergence (Caliendo et al., 2022; Prosser et al., 2024). Earlier epidemiological models primarily adopt mechanistic strategies based on biological assumptions, often structured around fixed compartments such as SIR or SEIR (Geng et al., 2021; Della Marca et al., 2023). While effective in simplified scenarios, they lack network topology and interaction semantics (Hunter & Kelleher, 2022) as they work under low-dimensional Ordinary Differential Equation systems that only track infected counts in discrete disease states, thus falling short in representing time-sequence reasoning and inter-location impact in real-world disease transmission.

To address the limitations of traditional epidemiological models, recent approaches have incorporated temporal architectures with Graph Neural Networks (GNNs) to capture spatio-temporal transmission dynamics from observational data (Liu et al., 2024). These models identify topological patterns both within and across time steps, enabling the learning of temporal correlations across spatially distributed locations in graph structures. This formulation offers a flexible and data-driven framework for supporting decision-making in disease surveillance and control, representing a significant advancement over classical forecasting methods (Brüel Gabrielsson, 2020; Liu et al., 2024). For example, Cola-GNN (Deng et al., 2020) captures the influence between the locations by combining the attention matrix with the geographical adjacency matrix. MSDNet (Tang et al., 2023) enhances regional epidemic predictions by integrating large-scale mobility data and fine-grained contact patterns.

Nevertheless, existing methods fail on two fronts. First, AIV forecasting formulation differs fundamentally from Influenza-Like Illness (ILI) forecasting. Conventional influenza-like and COVID-19 forecasting are typically modeled as *spatio-temporal processes* over locations, with transmission approximated via inter-location mobility flows and demographic factors. By contrast, AIV spread is inherently multi-source and multi-level. It spreads through *multiple concurrent pathways*, including long-range geographical seeding by migratory wild birds and evolutionary reassortment, operating on *distinct spatial and genetic levels*. This makes AIV forecasting a different formulation compared to ILI forecasting. Second, conventional modeling paradigm is mismatched to AIV dynamics. Previous models follow *a homogeneous setup*, where nodes represent locations and are connected with static adjacencies (Liu et al., 2023b; Yu et al., 2023; Lin et al., 2023) or learned correlations (Nguyen et al., 2023; Pu et al., 2024). This simplification assumes that *geographic proximity implies transmission risk*, focusing solely on location-level transmission and treating all infected cases as equally infectious. This paradigm overlooks subclade-specific transmission patterns and mutation-driven variation in virulence among cases, ignoring case-to-case interactions and temporal variability that often drive outbreak dynamics. Consequently, they struggle to capture multi-level transmission patterns, particularly how individual cases interact both within and across geographic regions.

Although recent efforts bring in meteorological variables (Lim et al., 2021; Papagiannopoulou et al., 2024) or integrate GNNs with mechanistic components (Cao et al., 2022; Wang et al., 2022a; Sha et al., 2021), they still operate under a homogeneous graph structure based on spatial locations. This limits their ability to model fine-grained case-level interactions critical to understanding the spread of AIV. Differently, genetic correlations can offer an epidemiologically informative view by capturing infection-driven linkages that are invisible to spatial or ecological proximity alone. By modeling cases and locations as distinct node types and constructing edges based on both spatial and genetic relations, the problem transfers from standard homogeneous GNNs to heterogeneous GNNs (HGNNs) (Zhang et al., 2019). HGNNs support multiple node and edge types, and are better equipped to handle multi-relational, multi-typed graphs, making them highly suitable for epidemiological modeling. Although prior works considering combining multi-type information (Hemker et al., 2024; Kim et al., 2023; Yu et al., 2022; Guo et al., 2023), they typically perform as a static multi-modal disease diagnosis for cases, ignoring the temporal patterns. These methods are not readily adaptable to the AIV forecasting context, where cases and locations each possess unique intra-layer connectivity patterns and changes over time, leading to spatio-temporal multi-layer graphs with changing node sets and distinct inter-layer semantics. Moreover, most existing fusion techniques do not explicitly preserve structural information during the fusion process, which can potentially lead to information loss in transmission structures. A principled information-preserving method is therefore essential to handle multi-layer heterogeneous graphs while preserving structural integrity and semantic distinctions across layers.

To this end, we systematically formulate AIV forecasting problem and design BLUE, a bi-layer heterogeneous graph fusion pipeline with dual layers that defines infectious cases and related locations as heterogeneous nodes within graphs, and integrates three types of information, e.g., spatial, genetic, and ecological information, into a unified framework for forecasting AIV outbreaks. The process begins by building a bi-layer heterogeneous graph that identifies diverse nodes and constructs multi-type edges. It then applies a cross-layer smoothing block inspired by Markov Random Fields (Dobruschin, 1968) to smooth heterogeneous connections. Trainable fusion nodes and fusion edges are formed to produce the fusion graph using a locality-sensitive hashing (LSH)-based sampler (Datar et al., 2004; Jafari et al., 2021) for efficient information integration. To preserve fine-grained structural semantics, we design a spectral regularizer that constrains the learned fusion graph to approximate that of the bi-layer structures, thereby maintaining its global diffusion geometry. Temporal dynamics are captured by an autoregressive framework, learning both spatial and temporal trends. To support evaluation and motivate further research, we publicly release a new avian influenza dataset, named **Avian-US** dataset[1]. Our main contributions are:

1. *New pipeline design*: We systematically define the AIV forecasting problem and propose **BLUE**, a pipeline that models heterogeneous nodes with multi-type information within a unified framework, establishing a principle formulation beyond prior work.

2. *Theoretical guarantees*: We design an information-preserving graph fusion to simplify the heterogeneous graphs without discarding the epidemiologically crucial structure, guaranteed with a theoretical bound from the spectral perspective.

---

[1]https://figshare.com/s/b369cd3447dd312ecd94, detailed in Appendix D.

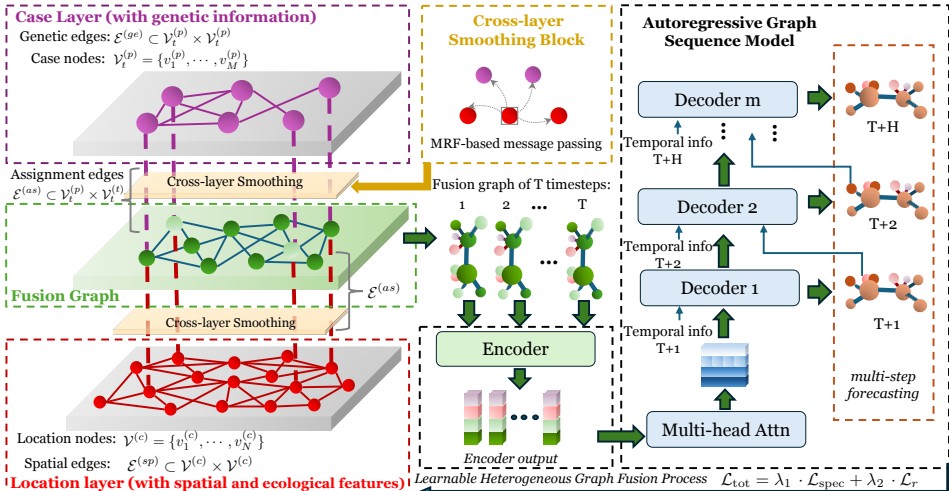

Figure 1: **BLUE** consists of 4 components: *Bi-layer Heterogeneous Graph Construction* models AIV spread using a bi-layer heterogeneous graph with two types of nodes (location and case) and three types of edges (spatial, genetic, and assignment). Then, the MRF-inspired *Cross-layer Smoothing block* aggregates neighbor information to create coherent representations for heterogeneous nodes and their connections. Graphs are then fused into *Information-Preserving Fusion Graphs* that preserve the original transmission structure using a spectral regularizer. Finally, *Autoregressive Encoder–Decoder Forecasting* encodes node interactions over time to generate multi-step forecasts.

3. *Empirical gains*: We publicly release and empirically validate **BLUE** on the avian influenza surveillance data in the United States, named **Avian-US**, demonstrating its superior performance.

## 2 RELATED WORKS

Based on the types of graph structures used, we summarize previous epidemiological methods into two categories: Static Graph-based (SG) approaches and Dynamic Graph-based (DG) approaches (refers to Appendix C for details). The topology of SG methods is fixed during training and inference, commonly specified by geographic adjacency or mobility priors (Xie et al., 2022; Yu et al., 2023; Lin et al., 2023; Liu et al., 2023b; Tang et al., 2023). Representative examples include EpiGNN (Xie et al., 2022), STEP (Yu et al., 2023), SMPNN (Lin et al., 2023), MSDNet (Tang et al., 2023), STSGT (Banerjee et al., 2022), and DGDI (Liu et al., 2023b). While these designs incorporate strong spatial priors, they cannot adjust the topology to reflect evolving or heterogeneous factors, such as case-to-case genetic relationships or changes, limiting expressiveness in AIV spread forecasting. For DG approaches, the topology evolves or is learned from data, allowing time-aware transmission patterns. Cola-GNN (Deng et al., 2020) implements a distance-based graph with attention to infer hidden cross-location dependencies. Epi-Cola-GNN (Liu et al., 2023a) couples SIR dynamics with a learnable, time-varying transmission matrix. CausalGNN (Wang et al., 2022a) introduces causal components to mitigate confounding and account for policy interventions. Despite flexibility, most DG models remain homogeneous and operate on a single location-view, overlooking heterogeneous relations that are central to characterizing multi-pathway transmission in AIV.

## 3 METHODOLOGY

**Problem Definition.** We formally define AIV forecasting as predicting future outbreaks by leveraging historical multi-source observations, structured as a sequence of heterogeneous graphs with multi-type relations. Let $\mathcal{G}_t = (\mathcal{T}_V, \mathcal{T}_E)$ denote a heterogeneous graph snapshot at week $t$, where $\mathcal{T}_V = \{\mathcal{V}^{(c)}, \mathcal{V}_t^{(p)}\}$ consists of location nodes $\mathcal{V}^{(c)}$ and case nodes $\mathcal{V}_t^{(p)}$. $\mathcal{T}_E = \{\mathcal{E}^{(sp)}, \mathcal{E}^{(ge)}, \mathcal{E}^{(as)}\}$ denotes *Spatial edges* (red lines), *Genetic edges* (purple lines), and *Assignment edges* (purple and red dash lines), respectively. *Spatial edges* $\mathcal{E}^{(sp)} = \{e_{<i,j>}^{(sp)}\}(i \in \mathcal{V}^{(c)}, j \in \mathcal{V}^{(c)})$ connects spatially neighboring locations from geographical modality. Each edge $\langle v_i^{(c)}, v_j^{(c)} \rangle$ is weighted by a geographic similarity score $\omega_{ij}^{sp}$ derived from inter-location distances. *Genetic edges* $\mathcal{E}^{(ge)} \subset \mathcal{V}_t^{(p)} \times \mathcal{V}_t^{(p)}$ connect genetically similar cases from biological modality, where each edge $\langle v_k^{(p)}, v_l^{(p)} \rangle$ carries a similarity

weight $w_{kl}^{(ge)}$ between case $k$ and case $l$. *Assignment edges* $\mathcal{E}^{(as)} \subset \mathcal{V}_t^{(p)} \times \mathcal{V}_t^{(t)}$ link each case to the location in which it was reported. Given a sequence of dynamic heterogeneous graphs over the past $T$ weeks, i.e., $\{\mathcal{G}_1, \mathcal{G}_2, \cdots, \mathcal{G}_T\}$, AIV forecasting is to predict the number of new infections in each location for the subsequent $H$ weeks $\{\mathbf{y}_{T+1}, \cdots, \mathbf{y}_{T+H}\} = f_\theta(\mathcal{G}_1, \cdots, \mathcal{G}_T)$, and our pipeline aims to learn the function $f_\theta$ that maps the historical graph sequence to future case counts accordingly.

**Bi-layer Heterogeneous Graph Construction.** Conventional GNN-based forecasting models for infectious diseases typically rely on a single-view representation of static spatial and temporal correlations among locations. These models assume that all infected cases contribute equally to transmission intensity, failing to distinguish between cases with high or low transmission potential. To overcome this limitation, we propose a bi-layer heterogeneous graph that simultaneously captures case-level infection intensity and location-level spatial connectivity within a single unified structure.

As shown in Fig.1, the bi-layer graph comprises two types of nodes, location nodes (red nodes) and case nodes (purple nodes), connected by three types of edges, *Assignment edges* (purple and red dash lines), *Genetic edges* (purple lines), and *Spatial edges* (red lines). A fixed set of **location nodes** $\mathcal{V}^{(c)} = \{v_1^{(c)}, \cdots, v_N^{(c)}\}$ represents $N$ distinct locations. Each location node $v_i^{(c)}$ is associated with a feature vector $\mathbf{x}_i^{(c)}(t) = [infected_i(t), population_i(t)]$, where $infected_i(t)$ is the number of newly reported infection cases and $population_i(t)$ represents the bird abundance in location $i$ at week $t$. Consistent with empirical studies showing localized avian influenza transmission (Bonney et al., 2018), we define spatial edges (red lines) between locations using a kernel-based weighting scheme rather than a rigid distance cutoff. For each undirected spatial edge $e_{<i,j>}^{(sp)}$ connecting locations $i$ and $j$, we assign a weight using a Gaussian kernel $\omega_{ij}^{sp} = K(D_{ij}) = \exp\left(-\frac{D_{ij}^2}{2\sigma^2}\right)$, where $D_{ij}$ is the geographic distance and $\sigma = \tau_d/3$ is the predefined connection range.

A time-varying set of **case nodes** $\mathcal{V}_t^{(p)} = \{v_1^{(p)}, \cdots, v_M^{(p)}\}$ represents individual infected samples reported at time $t$. The feature vector $\mathbf{x}_m^{(p)}(t)$ encodes the genetic profile of case $m$ at timestep $t$, computed as the average of its pairwise genetic distances to other cases using the Kimura 2-parameter (K80) model on aligned hemagglutinin (HA) segment sequences (Kimura, 1980). Genetic edges $\{e_{<m,n>}^{(ge)}\}$ connect cases $n$ and $m$ based on their genetic similarity, forming the case layer (see Appendix A for details). The final bi-layer heterogeneous graph is constructed by connecting each case node $v_m^{(p)}$ to its reported location node $v_i^{(c)}$ through an assignment edge $e_{<i,m>}^{(as)}$, thereby integrating the case and location layers.

### 3.1 Cross-layer Smoothing Block

Avian influenza transmission involves complex interactions across the case layer and location layer. We introduce the MRF-inspired cross-layer connection smoothing block to address the discrepancy of heterogeneous edges and nodes by explicitly leveraging local dependencies across heterogeneous graph neighborhoods, encouraging coherent representations within epidemiologically linked groups of nodes while preserving type-specific semantics.

The smoothing module leverages a mean-field approach that iteratively refines node embeddings on each heterogeneous graph. Formally, given an initial node embedding $x_v^{(0)}$ and three distinct edge types (spatial, genetic, and assignment edges), we perform $K$-time relation-specific message passing

$$m_r^{(k)}(v) = \frac{1}{|\mathcal{N}_r(v)|} \sum_{u \in \mathcal{N}_r(v)} \mathbf{W}_r x_u^{(k-1)} \tag{1}$$

where $\mathcal{N}_r(v)$ denotes immediate neighbors of node $v$. $W_r$ is a trainable parameter matrix representing the strength of interactions between connected nodes under relation $r$, adhering to the local Markov property. These messages, reflecting smoothed neighbor information, are aggregated across relations and combined with a node type-specific bias $b_{\tau(v)}$:

$$x_v^{(k)} = \text{ReLU}(\sum_r m_r^{(k)}(v) + \mathbf{b}_{\tau(v)}) \tag{2}$$

Here, we employ a ReLU activation, ensuring each node embedding is influenced by its neighbors' semantics. Iteratively applying this update $K$ times mimics multiple rounds of belief propagation, spreading information across the graph structure while explicitly considering different relational

contexts. Thus, by restricting propagation to immediate neighbors, applying learnable, relation-specific transformations, and iteratively refining node representations, this approach effectively integrates key aspects of MRF inference into a differentiable graph-based learning framework.

## 3.2 INFORMATION-PRESERVING FUSION GRAPHS

Heterogeneous graphs contain different types of nodes and relationships, making their structures inherently complex. This complexity arises from the simultaneous presence of diverse local and global relational dependencies. Converting heterogeneous graphs into a homogeneous form can significantly simplify representation; however, doing so typically risks losing valuable relational information from multiple sources. To overcome this limitation, we propose the *Fusion Graph* to transform the original heterogeneous structure, comprising location and case nodes connected by multiple relational types, into a unified one under spectral alignment (refers to Sec. 3.4 for details). Consequently, the Fusion Graph effectively captures the original heterogeneous complexity while providing a simpler and more interpretable structure (timestep $t$ is omitted for clarity).

**Fusion Nodes.** Implementationally, we define fusion nodes as aggregated representations of locations, constructed by systematically integrating neighbor information from the original heterogeneous graph. Given an initial heterogeneous graph $\mathcal{G}_t$, fusion node embedding $\mathbf{x}_i$ corresponding to the location node $v_i^{(c)} \in \mathcal{V}^{(c)}$ at timestep $t$ is generated by

$$\mathbf{z}_i^{(c)} = f_1\big(\mathbf{x}_i^{(c)} \,\|\, \mathbf{x}_i^{(\text{spatial})}\big), \ \mathbf{z}_i^{(p)} = f_2\big(\mathbf{x}_i^{(\text{genetic})}\big), \ \mathbf{x}_i = f_m\big([\mathbf{z}_i^{(c)} \,\|\, \mathbf{z}_i^{(p)}]\big), \tag{3}$$

where $\mathbf{x}\_i^{(c)}$ denotes the intrinsic features of location node $i$, and $\|\|$ represents the concatenation operation. The aggregation functions $f\_1$, $f\_2$, and $f\_m$ are implemented as MLPs with nonlinear activation functions. $\mathbf{x}_i^{(\text{spatial})}$ are the **spatial context vector** for location $i$, calculated by averaging over connected nodes of the same type $\mathbf{x}_i^{(\text{spatial})} = \frac{1}{|\mathcal{N}_i^{(\text{sp})}|} \sum_{j \in \mathcal{N}_i^{(\text{sp})}} \mathbf{x}_j^{(c)}$. $\mathbf{x}_i^{(\text{genetic})}$ is the **genetic context vector** for location $i$, computed by applying mean pooling to the feature vectors of all case nodes associated with location $i$. $\mathbf{x}_i^{(c\text{spatial})}$ aggregates information from neighboring locations connected by spatial relationships, while $\mathbf{x}_i^{(p\text{genetic})}$ summarizes genetic information derived from cases within location $i$. The specific aggregation functions, $f_1$ and $f_2$, and the subsequent fusion function $f_{(m)}$ employ Multi-Layer Perceptrons (MLPs) with nonlinear activation functions, effectively integrating diverse nodes into coherent fusion node embeddings.

**Fusion Edges.** Once fusion node embeddings are obtained, we construct edges to induce a coherent relational topology. Instead of forcing every possible pair or using a hard cut-off, we employ a learnable link prediction network augmented with Locality-Sensitive Hashing (LSH) to select edges and ensure computational tractability. Specifically, for any pair of fusion nodes $v_i$ and $v_j$ (generated from location nodes $v_i^{(c)}$ and $v_j^{(c)}$), we define the link probability $p_{ij}$ as $p_{ij} = \sigma(W_l[\mathbf{x}_i\|\mathbf{x}_j] + b_l)$, with $\sigma(\cdot)$ is Sigmoid activation for normalization probabilities. $\|$ indicate vector concatenatnion, and $W_l, b_j$ are learnable parameters. To avoid $O(N^2)$ enumeration over all node pairs, we generate a reduced candidate set via LSH: node embeddings are projected onto $K$ random hyperplanes to produce binary codes, which approximate cosine similarity in the embedding space (Charikar, 2002). Specifically, each fusion node embedding $\mathbf{x}_i^{(f)}$ is converted into a $B$-bit binary code

$$\mathbf{h}_i = [\text{sign}(\mathbf{r}_1^\top \mathbf{x}_i), \cdots, \text{sign}(\mathbf{r}_H^\top \mathbf{x}_i)] \in \{0, 1\}^B \tag{4}$$

where $\mathbf{r}_h$ are independent random projection vectors sampled from a spherical distribution and $B$ is the length of binary codes. Nodes with identical hash codes are placed into the same group, and within-bucket pairs $(i, j)$ are considered candidate edges, leveraging the high collision probability of similar vectors. If the number of exact-match candidates is below a predefined maximum $M_{max}$, the candidate set is supplemented by selecting node pairs whose Hamming distance between codes does not exceed a threshold $\tau_h$=1. By grouping nodes via code matches, we avoid the cost of exhaustive pairwise comparison and reduce to approximately $O(N + M_{max})$ operations, where $N = |\mathcal{V}^{(c)}| + |\mathcal{V}_t^{(p)}|$ denotes the total number of location and case nodes at time $t$.

To integrate original heterogeneous relations into the fusion graph, **BLUE** further employs a gate network that dynamically weights spatial and genetic edges based on node-pair interactions. Formally, for each pair of fusion nodes $v_i$ and $v_j$, the relation-specific embedding $\mathbf{e}_{ij}^{(r)} = \mathbf{e}_r + \mathbf{W}_{edge}\mathbf{x}_{ij}^{(r)} + \mathbf{b}_{edge}$, where $r \in \{spatial, genetic\}$, $\mathbf{e}_r$ is a learnable vector encoding relation type, and $\mathbf{x}_{ij}^{(r)} =$

$\{\mathbf{x}_i^{(c)}, \mathbf{x}_j^{(p)}\}$ contains the corresponding edge features. These embeddings are then fed into a multi-head self-attention network (Vaswani et al., 2017) to produce normalized scores across relations

$$\alpha_{ij}^{(r)} = \text{Attn}(\mathbf{x}_i, \mathbf{x}_j, \mathbf{e}_{ij}^{(r)}), \ \mathbf{e}_{ij}^{(r)} = \sum_r \frac{\exp(\alpha_{ij}^{(r)})}{\sum_{r'} \alpha_{ij}^{(r')}} \mathbf{e}_{ij}^{(r)} \tag{5}$$

The fusion-edge embedding is obtained as a weighted sum $\mathbf{e}_{ij} = \sum_r \alpha_{ij}^{(r)} \mathbf{e}_{ij}^{(r)}$. Thus, we obtain the unified fusion graphs with updated fusion node embeddings $\mathbf{X}_t$ and edge embeddings $\mathbf{E}_t^{(f)}$. This process enables our model to emphasize the most informative relationships for each node pair while preserving spatial and genetic interpretability and maintaining end-to-end differentiability. To ensure the maximum information preservation during the fusion process, we employ a spectral regularizer $\|\mathbf{L}_{hetero} - \mathbf{L}_f\|_F^2$ to ensure the diffusion modes consistency (detailed in Sec. 3.4).

### 3.3 Autoregressive Encoder-Decoder Forecasting

To model temporal dynamics over compressed fusion graphs, we use the sequence-to-sequence architecture. For each time step $t \in \{T - w + 1, \dots, T\}$, node embeddings $\mathbf{X}_t$ are processed by $L$ GraphSAGE layers $\mathrm{G}_l$ (Hamilton et al., 2017)

$$\mathbf{H}_t^{(l+1)} = \sigma\big(\mathrm{G}_l(\mathbf{H}_t^{(l)}, \mathbf{E}_t^{(f)})\big), l = 0, \dots, L-1, \tag{6}$$

where the initial layer input is the node embedding with an added positional encoding $\mathbf{H}_t^{(0)} = \mathbf{X}_t + \mathbf{p}_t$. Here, $\mathbf{p}_t$ is a learnable positional encoding. The outputs from the final GraphSAGE layer for each time step are stacked to form the overall representation $\mathcal{H} = [\mathbf{H}_{T-w+1}^{(L)}, \dots, \mathbf{H}_T^{(L)}] \in \mathbb{R}^{w \times N \times d}$. This tensor is then processed by a multi-head attention mechanism (Vaswani et al., 2017) to produce a context vector $\mathbf{H}^{(c)} = \text{Attn}(\mathcal{H})$, capturing the temporal dependencies. The decoder then forecasts for a horizon $H$, in an autoregressive manner. Final predictions are $\hat{\mathbf{y}}_h = \mathbf{W}_{out}\tilde{\mathbf{d}}_h$, where $\tilde{\mathbf{d}}_h$ is the decoder output. Further implementation and complexity details are in Appendix. A.

### 3.4 Optimization

During training, **BLUE** minimizes the objective that couples (i) multi-step forecasting term, (ii) spectral alignment of the learned fusion graph, and (iii) standard parameter regularization. Forecasting term measures the differences between the forecast $\hat{y}_{i,h}$ and the ground-truth infection count $y_{i,h}$. Since epidemiological data requires more critical predictions of higher infection cases than zero infection cases, we introduce a hierarchical weighting scheme that assigns different importance to prediction errors based on the magnitude of actual infection counts

$$\mathcal{L}_{pred} = \frac{1}{N} \sum_{h=T+1}^{T+H} \sum_{i=1}^{N} w_i \cdot (\hat{y}_{i,h} - y_{i,h})^2, \tag{7}$$

with three distinct infection severities $\tau_i$ ($i \in \{low, med, high\}$) with corresponding weight coefficients $w_i$. By emphasizing importance of higher infection cases with higher $w_i$, **BLUE** concentrates on non-zero infections through progressive weighting and learns to predict critical outbreak scenarios.

Compressing bi-layer graphs into fusion grpahs is a spectral low-pass filter: high-frequency components on the fine, case-level sub-graph are discarded. Consequently, operating only on the fused graph risks under-representing subtle transmission channels driven by a few genetically distinctive samples. To best preserve useful information, we employ a spectral regularizer to force the spectral alignment between the original bi-layer heterogeneous graphs and fusion graphs.

Let $P \in \{0,1\}^{(N+M_{max}) \times N}$ be the projection that maps case nodes into their home locations. $\mathbf{A}_{hetero}$ and $\mathbf{L}_{hetero}$ be the layer-adjacency and Laplacian of the heterogeneous graph, and let $\mathbf{A}_f$ and $\mathbf{L}_f$ be the adjacency and Laplacian of the learned fusion graph. Define the fusion Laplacian $\tilde{\mathbf{L}}_f := P\mathbf{L}_f P^\top$ and the corresponding normalized adjacencies $\mathbf{M}_{hetero} = \mathbf{I} - \mathbf{L}_{hetero}, \tilde{\mathbf{M}}_f = \mathbf{I} - \tilde{\mathbf{L}}_f$.

**Theorem 3.1.** *Assume spectral approximation under $\|\mathbf{L}_{hetero} - \tilde{\mathbf{L}}_f\| \le \varepsilon (0 < \varepsilon \ll 1)$. For any polynomial filter $p(\cdot)$ and content vector $\mathbf{H}$, we have $\|p(\mathbf{M}_{hetero})\mathbf{H} - p(\tilde{\mathbf{M}}_f)\mathbf{H}\|_F \le O(\varepsilon)\|\mathbf{H}\|_F$.*

*Proof.* Consider a GraphSAGE layer $\Phi(\mathbf{H}) = \sigma(\mathbf{W}_s\mathbf{H} + \mathbf{W}_n p(\mathbf{M})\mathbf{H})$, where weight norms $\|\mathbf{W}_s\| \le \beta_s, \|\mathbf{W}_n\| \le \beta_n$. Define $Z := \beta_s + \beta_n \|p(\mathbf{M}_{hetero})\|_2$. Under $\|\mathbf{L}_{hetero} - \tilde{\mathbf{L}}_f\|$,

$$\|\Phi_{hetero}(\mathbf{H}) - \Phi_f(P^\top\mathbf{H})\|_F \le O(\beta_n\varepsilon)\|\mathbf{H}\|_F + \underbrace{\beta_s\|\mathbf{H} - PP^\top\mathbf{H}\|_F}_{\text{fusion mismatch term}}. \tag{8}$$

Table 1: Performance on Avian-US dataset. Experiments are run under $T$=4 and $H$=4, with 5-fold validation. * denote statistically significant improvements, validated by a paired t-test at a significance level of $p < 0.05$ against the runner-up model. Best results are **bolded**.

| model | STGCN | SelfAttnRNN | ST-Net | EAST-Net | DRCNN | EpiGNN | Cola-GNN | STSGT | Epi-Cola-GNN | HGT | BLUE |
|---|---|---|---|---|---|---|---|---|---|---|---|
| RMSE($\downarrow$) | $0.874_{\pm0.143}$ | $0.896_{\pm0.175}$ | $0.902_{\pm0.160}$ | $0.897_{\pm0.177}$ | $0.797_{\pm0.131}$ | $0.783_{\pm0.120}$ | $0.712_{\pm0.100}$ | $0.826_{\pm0.125}$ | $0.691_{\pm0.114}$ | $\underline{0.652}_{\pm0.083}$ | $\mathbf{0.611}_{\pm\mathbf{0.072}}$ |
| MAE($\downarrow$) | $0.420_{\pm0.091}$ | $0.372_{\pm0.057}$ | $0.457_{\pm0.084}$ | $0.556_{\pm0.063}$ | $0.612_{\pm0.040}$ | $0.263_{\pm0.044}$ | $0.178_{\pm0.022}$ | $0.169_{\pm0.031}$ | $0.097_{\pm0.017}$ | $\underline{0.090}_{\pm0.016}$ | $\mathbf{0.085}_{\pm\mathbf{0.018}}$ |
| PCC($\uparrow$) | $0.048_{\pm0.002}$ | $0.052_{\pm0.005}$ | $0.058_{\pm0.003}$ | $0.065_{\pm0.004}$ | $0.059_{\pm0.005}$ | $0.072_{\pm0.005}$ | $0.077_{\pm0.004}$ | $0.040_{\pm0.008}$ | $\underline{0.078}_{\pm0.004}$ | $0.077_{\pm0.005}$ | $\mathbf{0.087}_{\pm\mathbf{0.013}}$ |
| SCC($\uparrow$) | $0.077_{\pm0.009}$ | $0.080_{\pm0.011}$ | $0.087_{\pm0.013}$ | $0.084_{\pm0.013}$ | $0.087_{\pm0.012}$ | $0.082_{\pm0.011}$ | $0.085_{\pm0.010}$ | $\underline{0.108}_{\pm0.020}$ | $0.083_{\pm0.011}$ | $0.097_{\pm0.017}$ | $\mathbf{0.122}_{\pm\mathbf{0.014}}$ |
| F1($\uparrow$) | $0.064_{\pm0.004}$ | $0.070_{\pm0.004}$ | $0.075_{\pm0.017}$ | $0.078_{\pm0.008}$ | $0.067_{\pm0.008}$ | $0.067_{\pm0.005}$ | $0.065_{\pm0.010}$ | $\underline{0.073}_{\pm0.018}$ | $0.082_{\pm0.020}$ | $0.080_{\pm0.008}$ | $\mathbf{0.100}_{\pm\mathbf{0.019}}$ |

Since heterogeneous nodes are fused at location level ($\mathbf{H} = PP^\top\mathbf{H}$), it simplifies to $\|\Phi(\mathbf{H}) - \Phi(P^\top\mathbf{H})\| \leq \beta_n\varepsilon\|\mathbf{H}\|$ without mismatch. **BLUE** are implemented with $L$ GraphSAGE layers,

$$\|\mathcal{F}_{hetero}(\mathbf{H}) - \mathcal{F}_f(P^\top\mathbf{H})\|_F \leq O(\frac{Z^L - 1}{Z - 1}\varepsilon)\|\mathbf{H}\|_F. \tag{9}$$

For a tighter upper bound, we apply the weight normalization on GraphSAGE layers to ensure $Z \approx 1$, reducing to an $O(L\varepsilon)$ bound. Hence, the fusion graph retains the effective information of the heterogeneous process up to a controllable, linear-in-depth spectral error. $\qquad\square$

With $\lambda_1, \lambda_2$ controlling the weights of spectral alignment and regularization, the final objective is

$$\mathcal{L}_{\text{tot}} = \underbrace{\frac{1}{N}\sum_{h=T+1}^{T+H}\sum_{i=1}^{N}w_i \cdot (\hat{y}_{i,h} - y_{i,h})^2}_{\mathcal{L}_{\text{pred}}} + \lambda_1 \cdot \underbrace{\left\|\mathbf{L}_{hetero} - \tilde{\mathbf{L}}_f\right\|_F^2}_{\mathcal{L}_{\text{spec}}} + \lambda_2 \underbrace{\sum_{w\in\Theta}\|w\|_2^2}_{\mathcal{L}_r}. \tag{10}$$

## 4 EXPERIMENTS

We evaluate **BLUE** on two datasets: (1) **Flu-Japan** dataset, introduced by (Deng et al., 2020), comprises weekly influenza-like illness counts from 47 prefectures in Japan over the period 2012–2019. (2) Our proposed **Avian-US** dataset, constructed as described in Appendix D, covers 3,227 counties and integrates genetic, spatial, and ecological modalities. Combining two datasets, we aim to demonstrate **BLUE**'s capacity to leverage both homogeneous and heterogeneous relational information[2].

**Baselines.** We compare **BLUE** against following GNN-based models: 1) homogeneous spatio-temporal forecasting models (**ST-GCN** (Yu et al., 2018), **SelfAttnRNN** (Cheng et al., 2016), **DCRNN** (Li et al., 2018), and **EAST-Net** (with a simplify version **ST-Net**) (Wang et al., 2022b)), 2) influenza-like epidemic prediction models (**Cola-GNN** (Deng et al., 2020), **EpiGNN** (Xie et al., 2022), **Epi-cola-GNN** (Liu et al., 2023a), **STSGT** (Banerjee et al., 2022)), and 3) heterogeneous-based model (**HGT** (Hu et al., 2020)). We adopt 5 complementary metrics: Root Mean Squared Error (**RMSE**), Mean Absolute Error (**MAE**), Pearson Correlation Coefficient (**PCC**), Spearman Correlation Coefficient (**SCC**), and Threshold F1 Score (**F1**, only detected when prediction exceeds threshold $t = 0.3$). Please refer to Appendix E for implementation details and experimental settings.

### 4.1 OVERALL PERFORMANCE

In the Avian-US dataset, **BLUE** achieves the highest PCC. However, as PCC is sensitive to small fluctuations and can be skewed in sparse datasets like Avian-US, we also incorporate SCC, a rank-based metric better suited for sparse, non-linear epidemic signals. As shown in Table.1, HGT is the runner-up in regression-based metrics (RMSE/MAE) owing to its capacity to model heterogeneous nodes. Epi-cola-GNN excels at capturing linear correlations, while STSGT demonstrates superior performance in outbreak detection, achieving the runner-up F1 score. Notably, **BLUE** achieves the highest PCC and SCC, confirming its strong capacity to capture both linear and non-linear correlations. Across all metrics, BLUE exhibits a clear advantage: it reduces the RMSE by 0.041 and the MSE by 0.005 compared to the runner-up, HGT. Furthermore, BLUE achieves the highest F1 score, which highlights its superior ability to detect outbreak occurrences rather than merely predicting precise case counts (comprehensive performance shown in Appendix. F).

### 4.2 PER-STEP PERFORMANCE

To provide finer-grained insights into forecasting performance, we conducted a temporal analysis of per-step results on the Avian-US dataset under $T = 4$ and $H \in \{4, 8\}$. As shown in Fig. 2, nearly

---

[2]Flu-Japan does not include genomic information for case–case correlations. Its graph structure reduces to only 47 prefecture-level homogeneous nodes connected by spatial edges. We include it as a fair comparison with previous baselines on homogeneous settings and move the detailed experiments on Flu-Japan to Appendix. F.

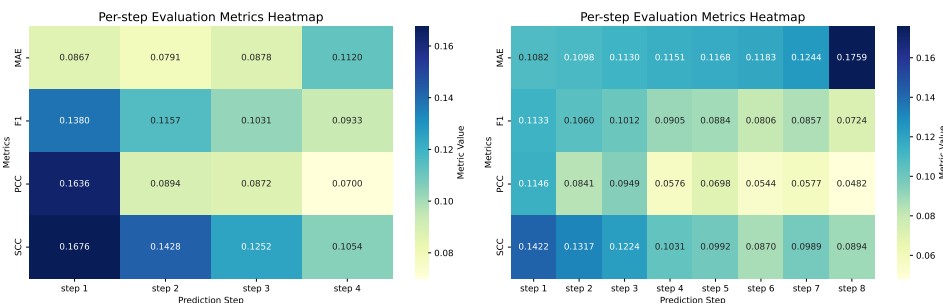

Figure 2: Per-step performance on Avian-US with $H = 4$ (left) and $H = 8$ (right).

all evaluation metrics decrease as the prediction horizon $H$ increases, which is consistent with the inherent difficulty of long-range forecasting. For per-step performance, **BLUE** performs optimally at step 1 under both horizon settings, achieving its highest F1 score, PCC, and SCC. Performance consistently declines as the predicted step increases, resulting in the lowest outbreak detection ability and largest prediction errors at the final step. Specifically, **BLUE**'s capacity to detect outbreaks decreases, with the F1 score falling from 0.1380 to 0.0933 for $H = 4$ and from 0.1133 to 0.0724 for $H = 8$. SCC, indicating the capacity for capturing spread trends, also decreases. Despite slight fluctuations, MAE of predicted infection counts increases with each successive step, rising from 0.0867 to 0.1120 for $H = 4$ and from 0.1082 to 0.1759 for $H = 8$.

### 4.3 ABLATION STUDY

Ablation studies are conducted to evaluate the contributions of each part in **BLUE** . We introduce following variants: 1) *w/o gen* only utilizes the spatial distances and assignment relationships; 2) *w/o CS+Spec* only implement the auto-regressive encoder-decoder framework; 3) *w/o CS* excludes the cross-layer smoothing block; 4) *w/o Spec* excludes the spectral regularizer from the objective function; 5) *w/ LSH* removes attention gate with simple averaging; 6) *w/ attn* only use attention gate for fusion edge calculation; 7)

Table 2: Ablation study on Avian-US.

| Variants | RMSE | MAE | F1 | PCC | SCC |
|---|---|---|---|---|---|
| w/o gen | 0.8093 | 0.1667 | 0.0721 | 0.0686 | 0.0946 |
| w/o CS | 0.7824 | 0.1828 | 0.0692 | 0.0677 | 0.0911 |
| w/o Spec | 0.8504 | 0.2310 | 0.0839 | 0.0729 | 0.1035 |
| w/o CS+Spec | 0.9020 | 0.2002 | 0.0605 | 0.0543 | 0.0869 |
| w/ LSH | 0.6141 | 0.1007 | 0.0994 | 0.0855 | 0.1105 |
| w/ attn | 0.6157 | 0.0945 | 0.0901 | 0.0755 | 0.1114 |
| w/o eco | 0.6772 | 0.1373 | 0.0978 | 0.0771 | 0.1093 |
| w/ drop | 0.6713 | 0.1254 | 0.0944 | 0.0801 | 0.1153 |
| BLUE | **0.6106** | **0.0848** | **0.1020** | **0.0871** | **0.1218** |

*w/o eco* removes location ecological features (bird abundance); 8) *w/ drop* randomly drop 20% of edges from bi-layer heterogeneous graphs. The results are shown in Table. 2.

The CS block and the Spectral Regularizer function work collaboratively. Disabling CS (*w/o CS*) allows noise from the initial sparse graph construction to propagate, harming ranking metrics. Similarly, removing the spectral constraint (*w/o Spec*) decouples the learned graph structure from the true epidemiological diffusion process, leading to overfitting. The combined removal of these components (*w/o CS+Spec*) yields the poorest performance, demonstrating that raw auto-regressive encoding is insufficient without structurally guided regularization. Removing genetic edges (*w/o gen*) causes a sharp increase in MAE, confirming that AIV outbreaks follow complex biological pathways that spatial proximity cannot fully explain. This is further supported by the *w/ drop* experiment, where randomly severing 20% of edges degrades performance, showing that AIV spread forecasting relies on a reliable, complete connectivity structure for aggregation. Moreover, two key observations arise from the attention experiments. First, replacing the dynamic gate with simple averaging (*w/ LSH*) increases MAE, proving that the model must learn to weigh spatial versus genetic risks adaptively. Second, using full dense attention (*w/ attn*) underperforms compared to the LSH-based approach (reducing F1 to 0.0901). This suggests that LSH acts as a beneficial constraint—limiting aggregation to high-probability neighborhoods prevents the model from overfitting to noisy, long-range interactions common in sparse data. Overall, **BLUE** achieves the best performance across all metrics, confirming that each component contributes distinctly to capturing the spatiotemporal dynamics of AIV spread.

### 4.4 SPECTRAL ALIGNMENT $\mathcal{L}_{spec}$

The spectral regularizer, $\mathcal{L}_{spec}$, ensures the alignment of transmission information between the original bi-layer heterogeneous graphs and the fused graphs. As demonstrated in Fig. 3, increasing the weight $\lambda_1$ steadily enhances the model's ability to capture trends, detect events, and reduce forecasting error. Specifically, as $\lambda_1$ increases across the tested range, RMSE shows a fluctuating

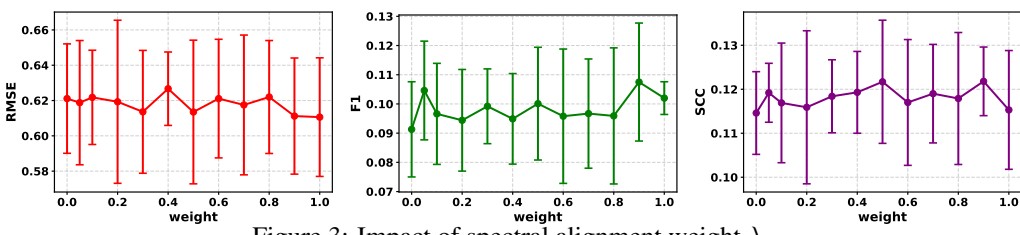

Figure 3: Impact of spectral alignment weight $\lambda_1$.

Table 3: Different predicted horizon $H$ under window size $T = 4$ (up) and $T = 8$ (down).

| $T$ | 4 | | | | $T$ | 8 | | | |
|---|---|---|---|---|---|---|---|---|---|
| $H$ | 1 | 2 | 4 | 8 | $H$ | 1 | 2 | 4 | 8 |
| RMSE | $0.6018 \pm 0.0730$ | $0.6098 \pm 0.0653$ | $0.6106 \pm 0.0719$ | $0.6174 \pm 0.0319$ | RMSE | $0.6337 \pm 0.0394$ | $0.6112 \pm 0.0411$ | $0.6268 \pm 0.0292$ | $0.6178 \pm 0.0434$ |
| MAE | $0.0955 \pm 0.0217$ | $0.0924 \pm 0.0198$ | $0.0848 \pm 0.0182$ | $0.0961 \pm 0.0358$ | MAE | $0.1151 \pm 0.0200$ | $0.0952 \pm 0.0245$ | $0.1061 \pm 0.0383$ | $0.1065 \pm 0.0212$ |
| F1 | $0.1271 \pm 0.0166$ | $0.1109 \pm 0.0273$ | $0.1001 \pm 0.0193$ | $0.0949 \pm 0.0113$ | F1 | $0.1226 \pm 0.0169$ | $0.1269 \pm 0.0175$ | $0.1030 \pm 0.0074$ | $0.0950 \pm 0.0069$ |
| PCC | $0.1195 \pm 0.0339$ | $0.0976 \pm 0.0188$ | $0.0871 \pm 0.0132$ | $0.0683 \pm 0.0074$ | PCC | $0.1225 \pm 0.0237$ | $0.1084 \pm 0.0151$ | $0.0828 \pm 0.0208$ | $0.0788 \pm 0.0088$ |
| SCC | $0.1489 \pm 0.0148$ | $0.1345 \pm 0.0230$ | $0.1217 \pm 0.0140$ | $0.1029 \pm 0.0066$ | SCC | $0.1587 \pm 0.0158$ | $0.1472 \pm 0.0085$ | $0.1208 \pm 0.0161$ | $0.1172 \pm 0.0127$ |

yet decreasing trend, from 0.6211 to 0.6112. The F1 Score rises from 0.0913 at $\lambda_1 = 0$ and peaks at $\lambda_1 = 0.9$, marking a 17.1% improvement over the baseline. Similarly, SCC steadily improves with the weight and reaches its maximum value at $\lambda_1 = 0.9$. Although a slight drop is observed at $\lambda_1 = 1$, the narrow error bars at this point suggest a more robust performance. Therefore, using an excessively large value for $\lambda_1$ could cause the model to over-prioritize spectral alignment, potentially leading to the neglect of the primary forecasting task.

### 4.5 IMPACTS OF WINDOW SIZE $T$ AND PREDICTED HORIZON $H$

**Impact of Prediction Horizon $H$.** When the observation window is short ($T = 4$), the lowest RMSE and MSE are achieved at the shortest prediction horizon. This suggests that **one-week patterns** in the Avian-US dataset are more stable and predictable than dynamics over longer horizons. As the prediction horizon $H$ increases, F1, PCC, and SCC tend to decline. This decline is attributed to the **high variability of temporal patterns** and the sparse ground-truth infection counts. The irregularity in avian influenza outbreaks, potentially due to variations in viral infectivity across different strains, makes long-term forecasting challenging. The worst results for all metrics are observed at the longest horizon ($H = 8$), a common outcome due to **error accumulation** in the autoregressive decoding process, indicating a loss of predictive accuracy when the forecast horizon is too long.

**Impact of Observation Window $T$.** Considering a fixed prediction horizon $H$, increasing the observation window $T$ (from $T = 4$ to $T = 8$) shows a consistent positive trend: (i) **Enhanced Pattern Capture**: There is a consistent decrease in RMSE and MAE, along with an upward trend in F1/PCC/SCC. This indicates that longer historical observations enhance BLUE's ability to capture underlying patterns in the Avian-US dataset. (ii) **Increased Variability**: the performance variances of nearly all metrics increase with larger $T$. This is likely because longer observation windows, especially with sparse data, introduce a higher proportion of zero-valued regions across counties, which enlarges variability across validation folds and leads to greater fold-to-fold variability in results.

## 5 CONCLUSION AND DISCUSSION

We present **BLUE**, a bi-layer heterogeneous graph fusion network that integrates genetic and spatial data to improve epidemic forecasting. **BLUE** uses cross-layer smoothing and information-preserving graph fusion to learn coherent representations of disease spread through an autoregressive encoder–decoder architecture. We evaluate **BLUE** on two datasets: the newly constructed Avian-US dataset and the publicly available Flu-Japan dataset. In both settings, **BLUE** outperforms strong spatio-temporal and epidemic forecasting baselines, demonstrating its effectiveness across different spatial and epidemiological contexts. The spectral fusion mechanism within **BLUE** is generalizable and can be extended to maintain structural alignment in more complex, multi-layer graph settings. While the current version of the Avian-US dataset does not include environmental variables, the architecture of **BLUE** is inherently modular and extensible. Additional information, such as location-level temperature, humidity, or other ecological indicators, can be incorporated as new features with corresponding cross-layer connections. In future work, we plan to expand the dataset with richer environmental attributes and biological categories, such as mammals and humans, to better support realistic forecasting, extending **BLUE**'s applicability to more comprehensive, real-world scenarios.

## 6 ETHICS STATEMENT

All authors have read the ICLR Code of Ethics and will adhere to it throughout submission, review, and discussion. Our study uses public, non-human data only; no human subjects or personally identifiable information are involved, and no animal experimentation is conducted. We follow the Code's principles to uphold scientific excellence with honest, transparent, and reproducible reporting; to respect privacy and confidentiality and comply with data licences; and to avoid harm and consider potential dual-use impacts. We disclose any conflicts of interest and funding transparently and welcome reviewer feedback on ethical concerns.

## 7 REPRODUCIBILITY STATEMENT

An anonymous code repository with training/inference scripts and configuration files is linked in the Abstract. Model architecture, forecasting procedure, and optimization objectives are specified in Section 3, including the autoregressive encoder–decoder framework (detailed in Appendix A.4) and loss/regularization details (in Section 3.4). Computational complexity and implementation notes are provided in Appendix A. Dataset sources, processing/alignment steps, and statistics are documented in Section 4 and Appendix D. The experimental setup, baseline implementations, evaluation metrics, and other practical details are summarized in Section 4 and Appendix E.

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

# A COMPUTATIONAL COMPLEXITY

Although **BLUE** introduces multiple components, each is carefully designed to ensure both scalability and practical implementability.

## A.1 BI-LAYER HETEROGENEOUS GRAPH CONSTRUCTION

At any given timestep $t$, the bi-layer heterogeneous graph contains $|V| = N + M_t$ nodes, where $N$ is the number of location nodes and $M_t$ is the number of infected case nodes at timestep $t$. When $\delta$ additional genetic samples are introduced, the node set grows to $|V| = N + M_t + \delta$. The corresponding edge sets expand: 1) Assignment edges: $E^{(as)}$: $O(M_t) \to O(M_t + \delta)$; 2) Genetic edges: $E^{(ge)}$: $O(kM_t) \to O(k(M_t + \delta))$ (assume a k-Nearest Neighbor connectivity). They result in a **linear increase** in graph construction complexity relative to the number of genetic samples.

## A.2 CROSS-LAYER SMOOTHING BLOCK

Given $K$-round smoothing, for all the relations $r \in \{sp, ge, as\}$, the cross-layer smoothing block calculates a relation-specific message passing process by averaging over the immediate neighbors $\mathcal{N}_r(v)$ and applying a learnable map $\mathbf{W}_r \in \mathbb{R}^{d \times d}$, then aggregates messages across relations with a type-specific bias and ReLU activation.

Let $|V| = N + M_t$ be the total number of nodes and $|\mathcal{E}_r|$ the number of edges of type $r$ (spatial, genetic, assignment). Each relation's neighbor aggregation corresponds to a sparse–dense product $\mathbf{D}^{-1}\mathbf{A}_r\mathbf{X}^{k-1}$, requiring $\mathcal{O}(|\mathcal{E}_r| \cdot d)$. The subsequent relation-specific transform $M_r^{(k)}\mathbf{W}_r$ costs $\mathcal{O}(|V| \cdot d^2)$ per relation $r$, and the element-wise nonlinearity is $\mathcal{O}(|V| \cdot d)$, which can be omitted.

Summing over the three relations and iterating K times yields $\mathcal{O}(K[d \cdot (|\mathcal{E}_{sp}| + |\mathcal{E}_{ge}| + |\mathcal{E}_{as}|) + 3|V| \cdot d^2])$. Since assignment edges link each infected case to its report location node and are binary, $|\mathcal{E}_{as}| = M$ in our setting.

## A.3 INFORMATION-PRESERVING FUSION GRAPHS

### A.3.1 FUSION NODES.

Fusion node embeddings are constructed by first aggregating neighboring locations (spatial) and within-location genetic information, and then employing MLPs $f_1$, $f_2$, $f_m$ to obtain the fused representations, which costs $\mathcal{O}(|\mathcal{E}_{sp}| \cdot d)$ for spatial neighbors and $\mathcal{O}(M \cdot d)$ for case-to-location assignment. The subsequent per-node MLP transforms cost $\mathcal{O}(N \cdot d)$.

### A.3.2 LOCAL SENSITIVE HASHING-BASED EDGE SELECTION.

**Code Construction.** For each bi-layer heterogeneous graph, the total number of location nodes is $N$. The length of each binary code for the Local Sensitive Hashing sampler is denoted as $B$. To generate LSH codes, we take $B$ dot products between node embeddings (with dimension $d$) and independently sampled random hyperplanes $\mathbf{r}_b$, retaining only the sign of each projection and resulting in a per-node cost of $O(Nd)$ for computing projections and $O(B)$ for extracting sign bits. Across all $N$ nodes, the overall cost is $O(NBd)$, primarily from matrix-vector multiplications. Once the LSH codes are computed, each node is inserted into a hash map using its binary signal as the key, adding a total $O(N)$ time cost for the hash table construction. The complete time complexity for code generation and hashing is thus $O(NBd + N)$.

**Bucket Matching.** Each bucket with $k$ nodes encodes $\frac{k(k-1)}{2}$ potential node pairs. The LSH-based sampler terminates when the number of collected fusion-edge pairs reaches the limit of $M_{max}$, ensuring this step remains within $O(M_{max})$ time.

If additional pairs are needed, we perform the following strategy: With randomly selected nodes, we search their neighbors within the Hamming distance threshold $\tau_h = 1$. Since the number of selected nodes and their neighbors are constants, the computation takes $O(B)$ in total.

Combining all terms gives:

$$T_{total} = O(NBd + N + M_{max} + B) \tag{11}$$

Since the embedding size $d$ and code length $B$ are fixed constants in implementation, the complexity simplifies to $O(N + M_{max})$, highlighting the scalability of our LSH-based sampler compared to naive pairwise similarity joins, which require $O(N^2)$ pairwise operations.

In practice, we replace the hash table implementation with a sorting-based alternative. Specifically, each node's $B$-bit binary code is converted into a single integer representation in $O(NB)$ time. The resulting integer codes are then sorted in $O(N \log N)$ time. Although the total time complexity of this sorting-based LSH variant becomes $O(N \log N + M_{max})$, which is asymptotically higher than the hash map-based approach, it is empirically faster on GPU architectures due to the inefficiency of hash table operations in parallel settings.

### A.3.3 FUSION EDGES.

Given a candidate set $M_{max}$ of fusion-edge pairs produced by LSH, fusion edges are then scored by the learnable link prediction network $p_{ij}$, requiring up to $\mathcal{O}(M_{max} \cdot d)$. The gate network then defines the relation-specific embeddings and computes normalized scores across relations. With a constant number of heads and relations, the complexity is mainly determined by per-pair projections, resulting in $\mathcal{O}(M_{max} \cdot d^2)$.

### A.4 ENCODER-DECODER FORECASTING

To model temporal dynamics, we adopt a sequence-to-sequence architecture over the compressed fusion graphs. At each time step $t$ of the window size $T$, the fusion node embeddings $\mathbf{X}_t$ are propagated through $L$ GraphSAGE layers $\mathbf{G}_l$ (Hamilton et al., 2017) [3]:

$$\mathbf{H}_t^{(l+1)} = \sigma(\mathbf{G}_l(\mathbf{H}_t^{(l)}, \mathbf{E}_t^{(f)})), \text{ for } l = 0, 1, \cdots, L-1 \tag{12}$$

where $\mathbf{H}_t^{(0)} = [\mathbf{X}_t + \mathbf{p}_t]$, $\mathbf{p}_t$ denotes learnable positional encoding. $\mathbf{E}_t^{(f)} = \{\mathbf{e}_{ij}\}$ is the edge embeddings of the fusion graph at timestep $t$. Over $T$ observations, we collect the final-layer outputs of each timestep $\{\mathbf{H}_t^{(L)}, \cdots, \mathbf{H}_{t+w-1}^{(L)}\}$ and stack them into $\mathcal{H} \in \mathbb{R}^{w \times N \times d}$. This tensor is fed into a temporal-aware fusion module, implemented by the multi-head attention network (Vaswani et al., 2017), to enforce features mutually concern across steps, capturing temporal dependence and yielding a context vector $\mathbf{H}^{(c)}$. Following, the decoder operates in an autoregressive manner over a forecasting horizon of length $H$. At each step $h \in \{1, \cdots, H\}$, it applies a GraphSAGE-based architecture composed of $L$ layers that mirror the encoder structure:

$$\mathbf{d}_h = \text{Decoder}(\mathbf{Z}_{h-1}^{(L)}, \mathbf{E}_T^{(f)}, \mathbf{f}_{h-1}, \mathbf{t}_h) \tag{13}$$

$$\mathbf{f}_{h-1} = W_{feat} \cdot (\mathbf{y}_{h-1}) + b_{feat} \tag{14}$$

Here, $\mathbf{Z}_{h-1}^{(L)}$ denotes the hidden state from the previous decoding step, and $\mathbf{E}_T^{(f)}$ represents the fusion edge embeddings at the final observation time $T$, and $\mathbf{t}_h$ encodes temporal information. $\mathbf{Z}_0^{(L)} = \mathbf{H}^{(c)}$. The decoded features $\mathbf{d}_h$ are then smoothly integrated with the previous step's decoded output $\mathbf{f}_{h-1}$ and the encoder's global context representation $\mathbf{H}^{(c)}$ through a weighted combination:

$$\tilde{\mathbf{d}}_h = (1 - \lambda_o - \lambda_p)\mathbf{d}_h + \lambda_o \mathbf{f}_{h-1} + \lambda_p \mathbf{H}^{(c)} \tag{15}$$

where $\lambda_o, \lambda_p \in [0, 1]$ control the reliance on current decoding, prior predictions, and global context, respectively, enhancing forecast stability across longer horizons. Finally, the prediction for step $h$, denoted as $\hat{\mathbf{y}}_h$, is generated via a nonlinear projection $\hat{y}_h = \mathbf{W}_{out}\tilde{\mathbf{d}}_h$. The hidden state for the current step is updated by a linear combination of $\tilde{\mathbf{d}}_h$ and the prior hidden states, enabling information flow across steps during sequential prediction.

**BLUE** performs forecasting over fusion nodes by aggregating representations from associated case nodes. While an increase of $\delta$ genetic samples adds more case nodes per location, **BLUE** utilizes pooling operations. These operations maintain a fixed dimensionality for node representations and do not expand the computational graph. Thus, the forecasting complexity depends solely on the number of location nodes $N$, remaining independent of $\delta$.

---

[3] We selected GraphSAGE as the backbone due to its balance between *theoretical compatibility* with the spectral regularization and *empirical performance* across datasets, detailed in Appendix B.

## A.5 SPECTRAL REGULARIZER

The spectral regularizer in **BLUE** preserves the diffusion geometry between the heterogeneous and fusion graphs by minimizing $\mathcal{L}_{\text{spec}} = \|\mathbf{L}_{\text{hetero}} - \tilde{\mathbf{L}}_f\|_F^2$, adding only a lightweight Frobenius-norm term between Laplacians. Importantly, we regularize only the top-$k$ eigenvalues, which capture the global diffusion modes. Computing these top-$k$ eigenpairs takes $O(k|V|^2)$ in theory but in practice is implemented with efficient iterative solvers that converge quickly due to the sparse structure of $\mathbf{L}_{hetero}$. Thus, the spectral regularizer adds negligible overhead relative to the main encoder–decoder forecasting.

## B  BACKBONE CHOICE

We selected GraphSAGE as the encoder-decoder in **BLUE** because it provides the best balance between *theoretical compatibility* with our spectral regularization framework and *empirical performance* across datasets.

**Theoretical justification.**  Our framework relies on a spectral regularizer to ensure that the fusion graph preserves the diffusion geometry of the original heterogeneous graph.

- **GCN:** The repeated application of symmetric normalized Laplacians in GCN accumulates spectral deviation linearly with depth, yielding an error up to $L\varepsilon$ after $L$ layers.

- **GraphSAGE:** Its inductive neighborhood aggregation achieves a tighter error bound, $\mathcal{O}\left(\frac{Z^L - 1}{Z - 1}\right) \cdot \varepsilon$, with $Z$ as the Lipschitz constant. Since we explicitly regularize $Z < 1$, the cumulative error remains controlled, making GraphSAGE theoretically well-aligned with our spectral regularization design.

- **GAT:** GAT redefines adjacency dynamically via feature-dependent attention. The resulting operator is **not governed by the regularized Laplacian**, and its evolving topology makes spectral consistency guarantees inapplicable.

**Empirical support.**  To validate our choice, we replaced GraphSAGE with GCN and GAT under identical settings ($L = 2$, spectral regularizer weight $= 0.2$, excluded for GAT). Results averaged over 5-fold validation and 5 random seeds are summarized below. GraphSAGE consistently outperforms alternatives on RMSE, MAE, and F1, while PCC differences are minor and consistent with PCC's fragility on sparse datasets (see Section 4.1).

Table 4: Performance comparison on Flu-Japan dataset.

|      | BLUE       | w/GCN  | w/GAT  |
| ---- | ---------- | ------ | ------ |
| RMSE | **1458**   | 1463   | 1595   |
| MAE  | **520**    | 732    | 675    |
| F1   | **0.8170** | 0.7908 | 0.7978 |
| PCC  | **0.7659** | 0.6733 | 0.6891 |
| SCC  | **0.5440** | 0.5022 | 0.5287 |

Table 5: Performance comparison on Avian-US dataset.

|      | BLUE       | w/GCN  | w/GAT  |
| ---- | ---------- | ------ | ------ |
| RMSE | **0.6106** | 0.6678 | 0.6791 |
| MAE  | **0.0848** | 0.0970 | 0.0942 |
| F1   | **0.1001** | 0.0967 | 0.0893 |
| PCC  | **0.0871** | 0.0664 | 0.0696 |
| SCC  | **0.1217** | 0.0931 | 0.1089 |

Both theoretically and empirically, GraphSAGE proves most compatible with **BLUE** 's information-preserving design. While PCC values remain low on Avian-US due to extreme sparsity, GraphSAGE yields clear improvements on RMSE, MAE, and F1, demonstrating superior suitability over GCN and GAT within our framework.

## C  RELATED WORKS

Based on the types of graph structures used, we summarize previous epidemiological methods into two categories: Static Graph-based (SG) approaches and Dynamic Graph-based (DG) approaches.

## C.1 STATIC GRAPH-BASED APPROACHES

SG methods rely on fixed graph structures throughout both training and prediction, typically incorporating predefined spatial or mobility-based priors such as geographic adjacency matrices (Xie et al., 2022; Yu et al., 2023; Lin et al., 2023) or static population flow matrices (Liu et al., 2023b; Tang et al., 2023). For example, EpiGNN (Xie et al., 2022) combines a static region-level graph with temporal modeling via spatio-temporal graph learning. STEP (Yu et al., 2023) and SMPNN (Lin et al., 2023) leverage graph neural networks to perform spatio-temporal forecasting on predefined location-level graphs constructed from geographic distances. MSDNet (Tang et al., 2023) defines the graph structure based on coarse-grained population migration trajectories and employs spatio-temporal graph learning to enhance prediction. Similarly, DGDI (Liu et al., 2023b) constructs geometric graphs derived from the location histories of infected individuals, implicitly modeling transmission potential via movement patterns. While effective for incorporating static spatial priors, they lack the flexibility to adapt to dynamic or heterogeneous factors, such as evolving case-to-case genetic relationships or ecological context, which limits their expressiveness in real-world epidemic spread.

## C.2 DYNAMIC GRAPH-BASED APPROACHES

DG approaches allow the graph's structure—either its edges, nodes, or both—to evolve over time or be updated through model-driven learning. This enables time-aware adaptation and more flexible representations of temporal transmission dynamics. For instance, Cola-GNN (Deng et al., 2020) begins with a static binary graph based on geographic distances but enhances it with a cross-location attention mechanism that learns hidden dependencies across regions. Epi-Cola-GNN (Liu et al., 2023a) builds on this by incorporating SIS dynamics and using a learnable transmission matrix to form time-varying graphs, better reflecting real-world epidemic progression. MepoGNN (Cao et al., 2022), on the other hand, explicitly integrates SIR dynamics into the graph learning process, allowing it to model evolving infectious connections more directly. CausalGNN (Wang et al., 2022a) introduces causal inference components to handle confounding effects and policy interventions, improving the reliability of predictions under complex real-world conditions. Despite their flexibility, these models still operate within a homogeneous framework, focusing on a single relational view. They overlook heterogeneous factors, such as genetic relationships, which are critical for understanding the multifaceted nature of real-world disease transmission.

Table 6: Dataset statistics.

| dataset | size (# locations $\times$ # week) | Min | Max | zeros (%) |
|---------|-----------------------------------|-----|-----|-----------|
| FLU-Japan | $47 \times 348$ | 0 | 26635 | 15.8 |
| Avian-US | $3227 \times 104$ | 0 | 92 | 98.4 |

# D    DATASET

The dataset statistics are shown in Table. D. The selection of the Flu-Japan and Avian-US datasets is strategic: they represent *two complementary epidemiological scenarios*, providing a rigorous test of **BLUE** under both moderate and extreme conditions.

**Flu-Japan (homogeneous baseline).**    Flu-Japan[4] is a well-established homogeneous benchmark dataset covering 47 prefectures. Its key strengths are:

- **Moderate spatial scale:** The relatively compact size enables interpretable analysis and clear visualization of epidemic dynamics.

- **Relatively continuous outbreaks:** Compared to Avian-US, Flu-Japan exhibits smoother temporal variation and less sparsity, making it well-suited to validate whether **BLUE** can capture epidemic *trends and temporal dynamics*.

Its main limitations are:

- **Limited spatial granularity:** Only 47 nodes, insufficient to stress-test scalability.

- **Single modality:** No genetic or environmental features, restricting evaluation of **BLUE** 's multi-modal integration capabilities.

**Avian-US (new large-scale dataset).**    To address these limitations, we introduce Avian-US, a new multimodal dataset covering 3,227 U.S. counties. It provides a far more challenging and realistic evaluation due to:

- **High spatial resolution:** Orders of magnitude more nodes than Flu-Japan, directly testing **BLUE** 's scalability.

- **Extreme sparsity:** Nearly 99% of weekly county-level series remain zero (Appendix C, Table 1), requiring robust handling of rare-event signals.

- **Multi-modal heterogeneity:** Incorporates spatial, genetic, and ecological data, capturing the complex drivers of avian influenza transmission in the U.S.

The main challenge is that sparsity and discontinuous transmission patterns make predictive correlation metrics fragile.

Taken together, Flu-Japan tests **BLUE** 's ability to learn epidemic trends under moderate scale and continuous outbreaks, while Avian-US stress-tests scalability, robustness, and multi-modal integration under sparse and heterogeneous conditions. This complementary pairing ensures that **BLUE** is validated under both interpretable benchmark settings and real-world large-scale challenges.

## D.1    AVIAN-US DATASET SETUP

The Avian-US dataset is a spatiotemporal, multi-modal dataset designed to support forecasting and modeling of avian influenza outbreaks across the United States. It integrates epidemiological records, viral genomic sequences, and host population data across 3,227 U.S. counties from 2021–2024. Each modality is spatially and temporally aligned at the county-week level, enabling multi-layered graph construction for downstream forecasting tasks.

---

[4]https://github.com/amy-deng/colagnn

## D.2 DATA COLLECTION

This dataset integrates spatiotemporal outbreak data, genomic sequences, and species-level abundance observations into a structured multilayer representation for disease forecasting. Each stream was independently collected but programmatically harmonized for modeling integration.

Infected case data were sourced from federal surveillance systems and include time-stamped infection reports for host data recorded at the county level across the continental United States from 2021 to 2024. Each record includes a free-text host descriptor, location metadata, and a collection date. To standardize taxonomic information, host descriptors were programmatically mapped to a reference taxonomy using a hierarchical classification schema derived from the International Ornithological Congress (IOC) avian taxonomy. This resolved inconsistencies such as overlapping or ambiguous common names by aligning to stable scientific identifiers.

Genomic data consist of hemagglutinin (HA) segment sequences found in publicly available viral genome repositories for Biotechnology Information (NCBI). Sequences with sufficient metadata were retained and filtered to include only wild bird hosts. A probabilistic record linkage model was used to associate sequences with case records. This model was trained on labelled match/non-match examples and used gradient-boosted decision trees to compute a match score based on taxonomic agreement, spatial proximity, and temporal overlap (within a ±14-day window). High-confidence matches were retained for downstream analysis.

Host population data were drawn from the eBird Status and Trends product, which provides weekly abundance estimates at 3 km resolution for North American bird species eBird (2022). Raster values were extracted for each species and week, then aggregated at the county level to align with the spatial granularity of case data. Only wild bird species were retained, and abundance vectors were indexed by county and week.

All records were assigned stable identifiers and organized into structured, timestamped tables. The pipeline ensures consistency across modalities while maintaining temporal fidelity and species-level resolution.

## D.3 DATA DESCRIPTION

The dataset comprises real-world, multi-source data documenting avian influenza outbreaks in the United States from January 2021 to December 2024. It includes temporally aligned information on confirmed infection cases, viral genome sequences, and wild bird abundance estimates, collected and harmonized at a weekly resolution.

The epidemiological component consists of over 12,000 reported H5-positive wild bird cases, spanning all 48 contiguous U.S. states. Each record includes collection date, geographic location (mapped to U.S. counties), and host classification. Taxonomic labels were normalised using a hierarchical mapping system that resolves ambiguous or underspecified entries to consistent species-level identifiers, informed by IOC naming conventions.

A subset of 8,000 cases was associated with full or partial HA segment sequences retrieved from public repositories. Genomic data were filtered to retain wild bird hosts only, and sequence metadata (host, date, location) were cleaned and harmonised to match epidemiological records. Genomic divergence between HA segment sequences was computed using the K80 model, which accounts for substitution asymmetry between transitions (A↔G, C↔T) and transversions. For each aligned sequence pair, we calculate the observed proportions of transitions ($P$) and transversions ($Q$), and estimate the evolutionary distance $d$ as:

$$d = -\tfrac{1}{2}\log(1 - 2P - Q) - \tfrac{1}{4}\log(1 - 2Q)$$

where $P = \frac{\#\text{transitions}}{L}$ and $Q = \frac{\#\text{transversions}}{L}$, with $L$ denoting the aligned sequence length. This evolutionary distance matrix encodes biologically grounded measures of divergence under a continuous-time Markov model and is well-suited for comparing within-clade avian influenza sequences. It is used as an input feature for constructing genomic similarity edges in the downstream heterogeneous graph.

Host population context was derived from over 630 weekly avian abundance layers produced by the eBird Status and Trends project eBird (2022). These layers estimate the relative abundance of bird

species at 3 km resolution across North America. Raster values were extracted and aggregated at the county level for all species matching wild bird families in the outbreak dataset. The resulting abundance vectors were aligned weekly to match case timelines and stored in compressed array format.

All data layers were temporally aligned by epidemiological week. Metadata were standardised across data types, with fields for date, location, taxonomic label, and abundance scores. Unique identifiers were assigned to all records to enable traceability across modalities. The dataset is designed to support temporal, ecological, and genetic analysis of avian influenza dynamics in wild bird populations using real-world observations, without reliance on synthetic or simulated data.

# E    IMPLEMENTATION DETAILS

In our empirical evaluations, we implement ST-GCN, SelfAttnRNN, DCRNN, and Cola-GNN using the open-source Cola-GNN repository [5]. ST-Net and EAST-Net are built upon the official EAST-Net implementation [6]. Implementation of Epi-GNN [7] and Epi-Cola-GNN [8] are based on their respective publicly available source code.

To enable evaluation of the proposed **BLUE** framework on the Flu-Japan dataset, we first construct the location layer based on the provided adjacency matrix. Each location node is assigned features based on the reported infection counts across prefectures. To represent case-level information, we simulate case nodes and associate them with their respective infected locations. Unlike the Avian-US setting, we assume uniform infectivity and assign an equal importance feature (value = 1) to each case node. This simplifies the transmission model by treating all cases as equally influential. With this setup, we construct the heterogeneous graphs for the Flu-Japan dataset and feed them into **BLUE** under the same modeling assumptions used in the Avian-US dataset.

## E.1    BASELINES.

We compare **BLUE** against following GNN-based models: 1) general spatio-temporal forecasting models (**ST-GCN** (Yu et al., 2018), **SelfAttnRNN** (Cheng et al., 2016), **DCRNN** (Li et al., 2018), and **EAST-Net** (with a simplify version **ST-Net**) (Wang et al., 2022b)), 2) homogeneous epidemic prediction models (**Cola-GNN** (Deng et al., 2020), **EpiGNN** (Xie et al., 2022), **Epi-cola-GNN** (Liu et al., 2023a), **STSGT** (Banerjee et al., 2022)), and 3) heterogeneous-based model (**HGT** (Hu et al., 2020)). Except HGT, all baselines are primarily designed for single-layer spatio-temporal forecasting and assume either a fixed or learnable graph structure. As such, they are not directly compatible with the multi-layer architecture of the Avian-US dataset. To make them applicable, we adapt each model by building homogeneous graphs tailored to its design. The adjacency matrix of STSGT is defined based on the geographical distance between locations. For ST-GCN, SelfAttnRNN, Cola-GNN, EpiGNN, EpiCola-GNN, and DCRNN, we define a binary adjacency matrix based on spatial proximity between counties. This setup mirrors their original use cases, allowing these models to learn spatio-temporal patterns over a fixed location-level graph. For ST-Net and EAST-Net, which support adaptive graph learning, we initialize homogeneous graphs with no predefined edges. These models learn the graph structure dynamically, allowing them to infer inter-location dependencies during training without relying on geographic priors.

## E.2    EXPERIMENTAL SETTINGS

To comprehensively evaluate the performance of the proposed method and baseline models, we adopt four complementary metrics: Root Mean Squared Error (**RMSE**), Mean Absolute Error (**MAE**), Pearson Correlation Coefficient (**PCC**), Spearman Correlation Coefficient (**SCC**)and F1 Score (**F1**). RMSE and MAE quantify the absolute and squared deviations between predicted and ground-truth counts. PCC measures the linear correlation between predicted and observed infection trends across spatial and temporal dimensions. F1 Score evaluates binary outbreak detection performance. To reflect this, We set a short observation window of $T$=4 steps and forecast the next $H$=4 steps (all experiments are conducted under this setting unless specified), and report the averaged evaluation metrics of $H$ steps. Experiments of all baselines and **BLUE** are conducted under a 5-fold cross-validation with the same random seed to ensure consistency. In addition to fixed embedding size $d$=8 and weight regularization $\lambda_2 = 5e - 4$, all baseline models are re-trained and tuned for optimal performance using their official open-source code. For **BLUE** , we search $\lambda_1 \in \{0.01, 0.05, 0.1, 0.5, 1\}$, and choose $\lambda_1 = 0.1$ for final evaluations. All experiments are run on either a single NVIDIA V100, DGX A100, or NVIDIA RTX A5000 GPU.

To ensure comparability in overall performance, we unify the embedding size and hidden dimensions across all models. We further apply the stratified weighting infection loss in Section 3.4 to all baselines

---

[5]https://github.com/amy-deng/colagnn

[6]https://github.com/underdoc-wang/EAST-Net/tree/main

[7]https://github.com/Xiefeng69/EpiGNN

[8]https://github.com/gigg1/CIKM2023EpiDL/tree/main

and re-run each baseline on the Avian-US dataset for fair comparison. The key hyperparameters used for all baseline models are listed below:

**ST-GCN**: embedding size=8, hidden dim=16, number of layers=3, epoch=100, learning rate=1e-5, dropout=0.3, window size=4, predicted horizon=4, weight of regularization term=5e-4.

**SelfAttnRNN**: embedding size=8, hidden dim=16, number of layers=2, epoch=100, learning rate=1e-5, dropout=0.3, window size=4, predicted horizon=4, weight of regularization term=5e-4.

**DCRNN**: embedding size=8, hidden dim=16, number of layers=2, max step of random walk=3, epoch=100, learning rate=1e-5, dropout=0.3, window size=4, predicted horizon=4, weight of regularization term=5e-4.

**Cola-GNN**: embedding size=8, hidden dim=16, number of filter=10, dilated rate for short term=1, dilated rate for long term=2, epoch=100, learning rate=1e-5, dropout=0.3, number of RNN layers=1, number of GNN layers=2, window size=4, predicted horizon=4, weight of regularization term=5e-4.

**ST-Net**: embedding size=8 (data) and 8 (time), Chebyshev layers=3, encoder layer=2, decoder layer=2, epoch=100, learning rate=1e-5, dropout=0.3, window size=4, predicted horizon=4, weight of regularization term=5e-4.

**EAST-Net**: spatial embedding size=8, modality embedding size =4, time embedding size=8, mobility prototype number = 8, memory dimension = 16, Chebyshev layers=3, encoder layer=2, decoder layer=2, epoch=100, learning rate=1e-5, dropout=0.3, window size=4, predicted horizon=4, weight of regularization term=5e-4.

**Epi-GNN**: embedding size=8, hidden dim=16, hidden dim of attention layer=64, pooling layer=2, patience=100, GNN layers=2, filer size=$\mathbf{f}_{1\times5,1}$ and $\mathbf{f}_{1\times3,1}$, window size=4, predicted horizon=4, epoch=100, learning rate=1e-5, dropout=0.3, weight of regularization term=5e-4.

**Epi-Cola-GNN**: embedding size=8, hidden dim=16, weight of epidemiological loss=0.5, patience=150, epoch=100, learning rate=1e-5, dropout=0.3, window size=4, predicted horizon=4, weight of regularization term=5e-4.

**STSGT**: embedding size=8, hidden dim=16, number of STSGT layers= 2, number of head= 2, dropout rate= 0.3, sampling number $n = 128$, sampling depth $L = 2$, learning rate=0.001, window size=4, predicted horizon=4, weight of regularization term=1e-4.

**HGT**: embedding size=8, hidden dim=16, number of layers= 2, dropout rate= 0.3, sampling number $n = 128$, sampling depth $L = 2$, learning rate=0.001, window size=4, predicted horizon=4, weight of regularization term=1e-4.

**BLUE**: embedding size=8, construction smoothing layer $K$=2, $B$=10, GraphSAGE layers $L$=2, $\lambda_1$=0.9, $\lambda_2$=5e-4, epoch=100, learning rate=1e-5, dropout=0.3, window size=4, predicted horizon=4. The infection severities are set as $\tau_{low} = 1$, $\tau_{med} = 5$, $\tau_{high} = 25$, the corresponding weight are $w_{low} = 1.0$, $w_{med} = 8.0$, $w_{high} = 15.0$. For the Avian-US dataset, we set $\lambda_o$=0.3 and $\lambda_p$=0.3. For the Flu-Japan dataset, we set $\lambda_o$=0.2, $\lambda_p$=0.3.

To manage data sparsity and numerical variation, we use four normalization techniques for preprocessing all feature data:

1. **Min-Max Normalization**: $x_{normalized} = (x - min)/(max - min)$. This method preserves value relationships but is sensitive to outliers.

2. **Z-Score Normalization**: $x_{normalized} = (x - mean)/std$. Suitable for approximately normally distributed features, it offers improved robustness to outliers compared to Min-Max scaling.

3. **Log-MinMax Normalization**: Applies a log transformation $log(x + \delta)$ followed by Min-Max scaling. This is effective for highly skewed positive-valued features with wide dynamic ranges.

4. **Log-Plus-One Normalization**: $log(x + 1)$. This transformation is appropriate for heavily skewed count data with many zeros, as it maintains the presence of zero values.

Log-based normalization is particularly effective for exponentially distributed features or datasets with a high proportion of zeros. Accordingly, we apply Log-Plus-One Normalization to the Avian-US dataset, which is highly skewed and sparsely populated. For the Flu-Japan dataset, which has more continuous variation and less sparsity, we apply Log-MinMax Normalization in the implementation.

### E.3 Spectral Regularizer Implementation

In our implementation, rather than enforcing strict equality between the Laplacians of the fusion graph and the original heterogeneous graph, we focus on preserving the most structurally informative components of the spectrum. Specifically, we constrain the largest $k$ eigenvalues of two graphs, which capture the global structures of the graph. By aligning the leading eigenvalues, we ensure that the fusion graph retains the essential global topology of the original heterogeneous graph and reduces sensitivity to discrepancies in the high-frequency components, which are often associated with local noise or unstable high-resolution variations. Consequently, the spectral loss remains robust and less prone to overfitting to noisy graph details.

Table 7: Overall performance on Flu-Japan dataset ($H = 4, T = 4$).

| Model | RMSE($\downarrow$) | MAE($\downarrow$) | PCC($\uparrow$) | SCC($\uparrow$) | F1($\uparrow$) |
|---|---|---|---|---|---|
| STGCN | 1763±764 | 995±726 | 0.7316±0.0794 | 0.4549±0.1345 | 0.6551±0.2437 |
| SelfAttnRNN | 1730±782 | 876±760 | 0.7850±0.0562 | 0.4138±0.0843 | 0.7040±0.2840 |
| ST-Net | 1761±727 | 1083±893 | 0.7898±0.0923 | 0.4193±0.1211 | 0.7171±0.2563 |
| EAST-Net | 1723±790 | 1038±848 | 0.7352±0.0573 | 0.4694±0.1185 | 0.6398±0.6637 |
| DRCNN | 1789±788 | 1216±1084 | 0.7674±0.0582 | 0.4104±0.0934 | 0.6663±0.2491 |
| EpiGNN | 1742±707 | 896±721 | 0.7401±0.0778 | 0.5029±0.1283 | 0.6699±0.2388 |
| Cola-GNN | 1600±765 | 886±758 | 0.7135±0.1054 | 0.4873±0.1632 | 0.6747±0.2472 |
| Epi-Cola-GNN | 1631±787 | 1368±1124 | 0.7486±0.0784 | 0.4908±0.1237 | 0.6831±0.2793 |
| STSGT | 1616±441 | 657±183 | 0.7493±0.1405 | 0.4281±0.0842 | 0.6782±0.2045 |
| HGT | 1527±476 | 565±193 | 0.5438±0.0725 | 0.3984±0.0914 | 0.6559±0.2757 |
| **BLUE** | **1511±331** | **553±150** | **0.7954±0.0328** | **0.5440±0.0586** | **0.7234±0.2420** |

# F EXPERIMENTAL RESULTS

## F.1 OVERALL PERFORMANCE

Prior methods are evaluated using long-term prior knowledge with $T = 20$ historical observations. However, such extended observation windows are impractical for real-world outbreak forecasting, where timely alerts are essential for intervention. To better reflect realistic forecasting constraints, we evaluate all baselines and **BLUE** using a shorter window of $T = 4$ (approximately one month) and forecast infection counts for future horizons $H = 4$ weeks, simulating rapid-response scenarios typical in epidemic modeling.

### F.1.1 FLU-JAPAN DATASET

As shown in Table 7, **BLUE** consistently outperforms all baselines in 5 evaluation metrics. It achieves the highest PCC and SCC, with notably stable performance compared to the greater fluctuations observed in baselines. This indicates that **BLUE** can capture the temporal dynamics of influenza outbreaks while reducing prediction errors simultaneously. In terms of outbreak detection, **BLUE** achieves the best F1 score. In all evaluation metrics, **BLUE** maintains competitive or lower standard deviations relative to baselines, indicating that its performance gains are consistent in short-term outbreak detection and prediction.

### F.1.2 AVIAN-US DATASET

In the Avian-US dataset, as shown in Table. 8, **BLUE** consistently achieves superior performance across error and correlation metrics. **BLUE** provides the highest F1 and lowest RMSE/MAE across all horizons, demonstrating its reliability in early-stage outbreak detection and accurate infectious number prediction. Different from results in Flu-Japan, **BLUE** shows lower PCC compared to SCC. As PCC captures linear correlation between predicted and ground-truth values, it is highly sensitive to such extreme values. This discrepancy can be attributed to the characteristical differences of two datasets, since Avian-US is highly sparse and contains several outliers while Flu-Japan is relatively continuous.

## F.2 IMPACTS OF WINDOW SIZE $T$ AND PREDICTED HORIZON $H$ ON FLU-JAPAN DATASET.

We evaluated **BLUE** under different observation sizes $T \in \{4, 8\}$ and predicted horizons $H \in \{1, 2, 4, 8\}$.

For each window size $T$, we compare the impact of the prediction horizon $H$ on the experimental results, shown in the Table.9. Notably, the F1 score improves from 53.73% to 72.34% when $T = 4$. The improvement indicates that multi-step decoding suppresses false alarms yet still captures infection patterns during several weeks. Single-week outbreak patterns are hard to predict accurately, while multi-week patterns are more likely to overlap with the actual disease outbreak's periodicity. However, the standard deviation of F1 increases with larger prediction horizon $H$, suggesting that outbreak

Table 8: Overall performance on Avian-US dataset ($H = 4, T = 4$).

| Model | RMSE($\downarrow$) | MAE($\downarrow$) | PCC($\uparrow$) | SCC($\uparrow$) | F1($\uparrow$) |
|---|---|---|---|---|---|
| STGCN | 0.8741±0.1428 | 0.4198±0.0911 | 0.0481±0.0024 | 0.0773±0.0087 | 0.0637±0.0043 |
| SelfAttnRNN | 0.8962±0.1752 | 0.3722±0.0566 | 0.0523±0.0052 | 0.0800±0.0111 | 0.0698±0.0038 |
| ST-Net | 0.9020±0.1603 | 0.4572±0.0840 | 0.0584±0.0034 | 0.0865±0.0128 | 0.0751±0.0167 |
| EAST-Net | 0.8973±0.1772 | 0.5556±0.0626 | 0.0647±0.0037 | 0.0839±0.0134 | 0.0779±0.0083 |
| DRCNN | 0.7965±0.1314 | 0.6123±0.0395 | 0.0588±0.0053 | 0.0871±0.0117 | 0.0665±0.0078 |
| EpiGNN | 0.7834±0.1200 | 0.2627±0.0440 | 0.0719±0.0049 | 0.0816±0.0112 | 0.0672±0.0053 |
| Cola-GNN | 0.7118±0.1001 | 0.1777±0.0223 | 0.0770±0.0042 | 0.0846±0.0102 | 0.0653±0.0096 |
| Epi-Cola-GNN | 0.8264±0.1246 | 0.0967±0.0172 | 0.0780±0.0042 | 0.0825±0.0112 | 0.0816±0.0197 |
| STSGT | 0.6907±0.1143 | 0.1690±0.0312 | 0.0403±0.0080 | 0.1082±0.0201 | 0.0725±0.0183 |
| HGT | 0.6523±0.0828 | 0.0902±0.0163 | 0.0772±0.0054 | 0.0966±0.0165 | 0.0801±0.0077 |
| **BLUE** | **0.6106 ± 0.0719** | **0.0848 ± 0.0182** | **0.0871 ± 0.0132** | **0.1217 ± 0.0140** | **0.1001 ± 0.0193** |

Table 9: Different predicted horizon $H$ under window size $T = 4$ (up) and $T = 8$ (down) on the Flu-Japan dataset. We report the experimental results with $\lambda_p = 0.3$ and $\lambda_o = 0.2$.

| $T$ | 4 | | |
|---|---|---|---|
| $H$ | 1 | 2 | 4 |
| RMSE | 1486.3070 ± 457.1804 | 1485.2236 ± 443.3038 | 1511.3080 ± 331.2720 |
| MAE | 556.4849 ± 236.3284 | 555.2262 ± 222.3261 | 553.1024 ± 150.2401 |
| F1 | 0.5373 ± 0.0397 | 0.6763 ± 0.0114 | 0.7234±0.2420 |
| PCC | 0.6301 ± 0.0221 | 0.6317 ± 0.0927 | 0.7954±0.0328 |

| $T$ | 8 | | | |
|---|---|---|---|---|
| $H$ | 1 | 2 | 4 | 8 |
| RMSE | 1528.6636 ± 335.5966 | 1529.5472 ± 362.8420 | 1524.5787 ± 359.4490 | 1523.6165 ± 382.6201 |
| MAE | 564.5333 ± 176.0671 | 569.2224 ± 186.9960 | 565.1868 ± 184.9081 | 572.1089 ± 207.6871 |
| F1 | 0.6471 ± 0.3699 | 0.7353 ± 0.3678 | 0.7568 ± 0.3547 | 0.7321 ± 0.3662 |
| PCC | 0.6710 ± 0.0252 | 0.5732 ± 0.0930 | 0.5536 ± 0.1643 | 0.5233 ± 0.2590 |

detection becomes more variable and challenging across validation folds as the forecasting window extends. In contrast, RMSE and MAE gradually decrease as $H$ increases, implying that the Flu-Japan dataset exhibits more consistent multi-week outbreak patterns, making longer-horizon forecasts easier to stabilize. In comparison, short-term trends may exhibit higher fluctuations, which may be less structured. When $T = 8$, RMSE varies $\leq 0.35\%$ and MAE remains within $1\%$ across all horizons, illustrating that the model has sufficient context to predict each county's epidemic with 8-week observations. F1 peaks at $75.68\%$ for a four-week horizon, further proving our finding of multi-week patterns under the $T = 4$ setting and highlighting the effectiveness of **BLUE** in outbreak detection when sufficient prior context is available.

For a fixed prediction horizon $H$, we observe that F1 consistently improves as the observation window $T$ increases. This indicates that a longer historical context enhances the model's ability to distinguish outbreak weeks from background fluctuations, thereby improving the confidence and precision of binary outbreak detection. Conversely, we find that RMSE and MAE increase with larger $T$, although their error bars become tighter. This indicates that longer observation windows enhance the consistency of predictions, but may come at the cost of reduced alignment with recent temporal patterns.

### F.3 PER-STEP RESULTS

To provide finer-grained insight into forecasting performance over time, we conducted a temporal analysis of per-step results for both Flu-Japan and Avian-US, under the setting $T = 4$, $H = 4$, comparing **BLUE** against Epi-Cola-GNN. SCC (Spearman correlation coefficient) is also reported to capture rank-based trend consistency.

**Per-step performance.** Tables 10 and 11 show detailed metrics at each forecast horizon. Generally, all models exhibit gradual performance decline with increasing horizon, consistent with the inherent difficulty of long-horizon forecasting.

Table 10: Per-step results on Flu-Japan.

| BLUE | RMSE | MAE | F1 | PCC | SCC |
|---|---|---|---|---|---|
| Step 1 | 1170.0736 | 379.1464 | 0.7998 | 0.7819 | 0.5389 |
| Step 2 | 1360.5968 | 431.5831 | 0.8069 | 0.7789 | 0.4440 |
| Step 3 | 1601.2518 | 511.6768 | 0.8132 | 0.6966 | 0.3286 |
| Step 4 | 1562.1200 | 517.4587 | 0.8186 | 0.6217 | 0.2231 |
| Horizon avg. | 1423.5106 | 459.9625 | 0.8096 | 0.7197 | 0.3836 |
| Epi-Cola-GNN | RMSE | MAE | F1 | PCC | SCC |
| Step 1 | 573.1535 | 352.1775 | 0.7959 | 0.7138 | 0.4620 |
| Step 2 | 1415.7948 | 842.4990 | 0.7811 | 0.7965 | 0.3413 |
| Step 3 | 1914.2330 | 1171.5552 | 0.7141 | 0.6239 | 0.1458 |
| Step 4 | 2508.5769 | 1707.4314 | 0.4093 | 0.6589 | 0.2321 |
| Horizon avg. | 1602.9395 | 1018.4157 | 0.6751 | 0.6982 | 0.2953 |

Table 11: Per-step results on Avian-US.

| BLUE | RMSE | MAE | F1 | PCC | SCC |
|---|---|---|---|---|---|
| Step 1 | 0.6154 | 0.1120 | 0.1380 | 0.1636 | 0.1676 |
| Step 2 | 0.6467 | 0.0878 | 0.1157 | 0.0894 | 0.1428 |
| Step 3 | 0.4691 | 0.0791 | 0.1031 | 0.0872 | 0.1252 |
| Step 4 | 0.7521 | 0.0867 | 0.0933 | 0.0700 | 0.1054 |
| horizon avg. | 0.6208 | 0.0914 | 0.1125 | 0.1025 | 0.1352 |
| Epi-Cola-GNN | RMSE | MAE | F1 | PCC | SCC |
| Step 1 | 0.6826 | 0.0877 | 0.0846 | 0.0792 | 0.1086 |
| Step 2 | 0.7544 | 0.1294 | 0.0827 | 0.0754 | 0.0823 |
| Step 3 | 0.5765 | 0.0953 | 0.0832 | 0.0756 | 0.0735 |
| Step 4 | 0.6733 | 0.1042 | 0.0881 | 0.0724 | 0.0569 |
| Horizon avg. | 0.6716 | 0.1042 | 0.0845 | 0.0757 | 0.0803 |

**Findings.** On Flu-Japan, `BLUE` shows consistently robust performance across all forecast steps. While Epi-Cola-GNN performs marginally better at Step 1, its F1 score declines thereafter. In contrast, `BLUE` degrades more gracefully, achieving higher average F1, PCC, and SCC, demonstrating stable trend capture over time.

On Avian-US, despite extreme sparsity, `BLUE` attains lower horizon-averaged RMSE and MAE and consistently higher F1 than Epi-Cola-GNN. While PCC remains marginally lower (a metric noted for its fragility under sparsity), `BLUE` achieves substantially higher SCC, confirming its superior ability to preserve nonlinear temporal ranking patterns critical for outbreak detection.

These per-step analyses demonstrate that `BLUE` not only improves average error metrics but also captures temporal patterns more reliably, degrading gracefully over longer horizons. We will include comprehensive per-step results for all baselines (including SCC) in the final version for completeness.

### F.4 IMPACT OF $\lambda_o$

In the decoder process, $\lambda_o$ controls the influence of the previous step's decoded output $\mathbf{f}_{h-1}$. To investigate its effect, we fix $T = 4$, $H = 4$, $\lambda_p = 0.3$ (the weight of the encoder's hidden state $\mathbf{H}^{(c)}$) and only vary $\lambda_o$ in $\{0.1, 0.2, 0.3, 0.4, 0.5, 0.6\}$. Lower values of $\lambda_o$ emphasize the decoded feature of the current step, while higher values increase the model's reliance on earlier decoder outputs. The results are summarized in Table.12.

On the Flu-Japan dataset, we observe that setting $\lambda_o = 0.5$ achieves the lowest RMSE and MAE, while $\lambda_o = 0.2$ achieves the highest PCC. PCC initially increases with rising $\lambda_o$ before declining, which shows an opposite trend compared to F1 score, suggesting a trade-off between temporal consistency and outbreak detection accuracy. A similar pattern appears in RMSE and MAE, which first worsen and then improve as $\lambda_o$ increases. indicating that a balanced amount of information from previous decoding steps helps the model better forecast the temporal progression of disease, while too little or too much may constrain its ability to generalize effectively.

Table 12: impact of $\lambda_o$ on the Flu-Japan dataset (up) and the Avian-US dataset (down).

| $\lambda_o$ | RMSE | MAE | F1 | PCC | SCC |
|---|---|---|---|---|---|
| 0.1 | 1488.3113 | 537.1610 | 0.9174 | 0.6742 | 0.5218 |
| 0.2 | 1511.708 | 553.9024 | 0.7234 | 0.7954 | 0.5440 |
| 0.3 | 1494.3136 | 540.6177 | 0.7310 | 0.6303 | 0.5010 |
| 0.4 | 1476.6810 | 528.6474 | 0.7959 | 0.6285 | 0.5392 |
| 0.5 | 1472.6313 | 526.4264 | 0.9172 | 0.6630 | 0.4938 |
| 0.6 | 1478.7874 | 528.7906 | 0.9165 | 0.6566 | 0.4211 |

| $\lambda_o$ | RMSE | MAE | F1 | PCC | SCC |
|---|---|---|---|---|---|
| 0.1 | 0.6255 | 0.0549 | 0.0268 | 0.0074 | 0.6742 |
| 0.2 | 0.6288 | 0.0653 | 0.0221 | 0.0179 | 0.6742 |
| 0.3 | 0.6178 | 0.0665 | 0.0265 | 0.0241 | 0.6742 |
| 0.4 | 0.5295 | 0.0678 | 0.0222 | 0.0122 | 0.6742 |
| 0.5 | 0.6181 | 0.1068 | 0.0213 | 0.0313 | 0.6742 |
| 0.6 | 0.6264 | 0.1776 | 0.0213 | 0.0480 | 0.6742 |

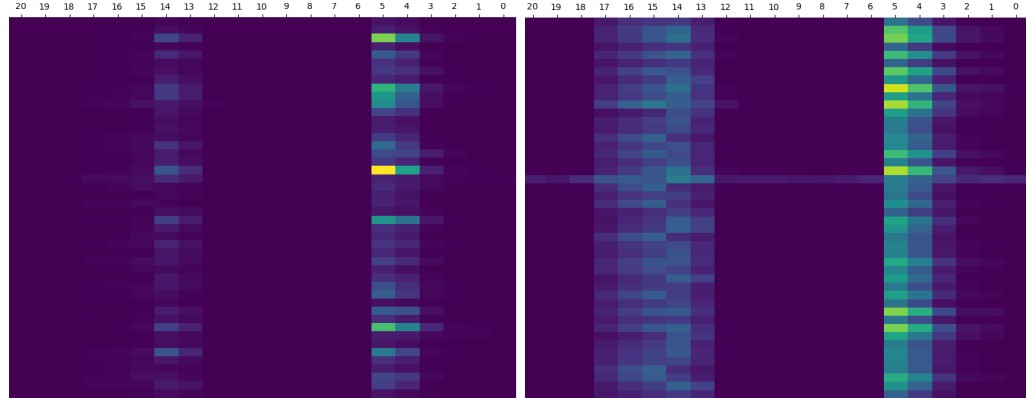

Figure 4: Heatmap of ground truth (left) and prediction (right) on Flu-Japan dataset.

In the Avian-US dataset, the F1 score and MAE peak at $\lambda_o = 0.1$, suggesting that leveraging current-step signals is more effective for identifying sparse outbreaks. With the increase of $\lambda_o$, MAE and F1 score gradually decrease with PCC significantly increase, suggesting that emphasizing previous predictions may reduce the sensitivity to sudden outbreak events and accurate infectious forecasting, but can enhance trend alignment by capturing long-horizon patterns.

### F.5 CASE STUDY ON FLU-JAPAN DATASET

We provide the heatmap of experimental results on the Flu-Japan dataset, shown in Fig. 4. Both the ground truth and predicted heatmaps exhibit a distinct peak in activity during the 8th batch. **BLUE** not only identifies this peak at the correct temporal position but also accurately reproduces its spatial distribution across multiple prefectures, demonstrating that the model has effectively learned both the timing and spatial structure of the outbreak. In the ground truth heatmap, bright horizontal bands correspond to a small subset of highly infected prefectures. **BLUE** 's predictions highlight these same regions while maintaining low activity for the remaining prefectures, indicating that the model captures the dominant transmission pathways without overpredicting infection spread.

## G THE USE OF LARGE LANGUAGE MODELS

LLMs were used exclusively for the editorial refinement of the manuscript text, focusing on grammar, wording, and overall clarity. They were not employed to generate scientific content, design

experiments, write code, or produce research results. All authors have reviewed the final content and assume full responsibility for it, in alignment with the ICLR's LLM usage guidance and Code of Ethics.

