# OpenReview forum: "BLUE: Bi-layer Heterogeneous Graph Fusion Network for Avian Influenza Forecasting"
_ICLR.cc/2026/Conference — ICLR 2026 Conference Withdrawn Submission_

### Official Review · Reviewer_kX83 · 2025-11-02

**Soundness:** 2
**Presentation:** 3
**Contribution:** 1
**Rating:** 2
**Confidence:** 4

**Summary:**

This paper presents BLUE, a bi-layer heterogeneous graph fusion network for avian influenza (AIV)
outbreak forecasting. The method models both spatial and genetic relationships through two graph
layers, a location layer (geographical and ecological data) and a case layer (genetic data of virus samples).
A cross-layer smoothing block (inspired by MRFs) and an information-preserving graph fusion module
align heterogeneous information into a unified structure, optimized with a spectral regularizer ensuring
diffusion consistency. An autoregressive encoder–decoder then predicts future outbreaks. The authors
also introduce a new Avian-US dataset combining genetic, spatial, and ecological modalities. Experiments
show BLUE outperforms strong baselines (e.g., HGT, Cola-GNN, EpiGNN) on both Avian-US and Flu-Japan
datasets.

**Strengths:**

The paper tackles a scientifically meaningful and socially important problem with clear relevance to both
epidemiology and AI for science. The bi-layer heterogeneous graph design and spectral information-
preserving fusion are innovative and well-motivated. The theoretical analysis (Theorem 3.1) is a strong
addition. Combining spatial, genetic, and ecological information within a principled heterogeneous
framework is technically elegant. Besides, the dataset creation, which integrates genomic and spatial
ecological sources, is a valuable community contribution.

**Weaknesses:**

1. The model complexity (bi-layer + smoothing + fusion + autoregressive decoder) may be excessive for
datasets with limited sample size. The paper could benefit from a complexity–performance tradeoff
analysis. And the generalization analysis is limited in the paper. While the method is compelling for avian
influenza, the paper lacks experiments or discussion on whether the approach generalizes to other
diseases or epidemic structures (e.g., human COVID-19, plant viruses).
2. The method is lack of novelty. The design and usage of heterogeneous gnn is standard. The
authors simply leverage it to solve a domain problem. I don't see novelty for the technical part.
1. where does the and in Eq. 3 come from?
2. Line 216, it seems that that the message-passing mechanism is mainly used to ensure "each node
embedding is influenced by its neighbors' semantics".

**Questions:**

see above.

---

> ### Author Response · Authors · 2025-11-24
> **R1. Complexity and Generalizability of BLUE**
>
> We thank the reviewer for the valuable concerns. We address the concerns regarding complexity and generalization by demonstrating that BLUE’s design offers a superior computational trade-off and a flexible topology for other pathogens from the following aspects.
>
> **1. Complexity-Performance Trade-off Analysis**
>
> Contrary to the perception of "excessive" complexity, BLUE is more efficient than standard baselines while delivering necessary performance gains.
>
> - **BLUE (per encoder window $w$ and horizon $H$)**: Let $N$ be the number of location nodes. $M_{\max}(t)$ be the maximum number of fusion candidates at time $t$ and $M_{\max}^*=\max_t M_{\max}(t)$ is the upper-bound of case counts. With fixed hidden widths and layer counts (constants $d,B,K,L$ in our experiments) and sparse graphs ($|E^{\mathrm{sp}}|=O(N)$, $|E_t^{\mathrm{ge}}|=O(kM_t)$, $|E_t^{\mathrm{f}}|=O(M_{\max}(t))$), the total cost is:
>
> $$
> C_{\text{BLUE}} = O\left( \sum_t (N+M_t+M_{\max}(t)) + L(w+H)(N+M_{\max}^*) + N w^2 \right).
> $$
>
> The three terms correspond to the costs of **fusion graph construction**, **GNN inference over fused graphs**, and **temporal attention**, respectively.
> The fusion stage leverages **LSH** for the $O(N\log N)$ neighborhood search (detailed in **Appendix A.3.2, page 14–15**). The subsequent GraphSAGE forecasting operates on the fused location graph and is **independent of $\sum_t M_t$**.
>
> - **Cola-GNN (for comparison)**
>
> With RNN hidden size $D$, attention size $d_a$, $K_f$ dilated-conv filters, $L$ GNN layers, and per-layer width $F$, the estimated complexity is:
>
> $$
> \mathcal{T}_{\text{Cola-GNN}} = O\Big( N T D^2 + N K_f T^2 + (N d_a D + N^2 d_a + N^3) + L(N^2 F + N F^2) \Big).
> $$
>
> The four terms represent the costs of **RNN**, **dilated convolution**, **attention/gating mechanisms**, and **message passing**, respectively. Hence, for large $N$, **BLUE’s sequence component scales near-linearly in $N$**, while key terms in Cola-GNN are **quadratic or even cubic** in $N$. Empirically, **BLUE** translates to better accuracy–cost tradeoffs **at higher spatial resolution**.
> Therefore, for our large-scale graph ($N=3,227$), BLUE is computationally lighter than these "simpler" baselines.
>
> - **Performance Gain**: AIV forecasting is the **minimal structure** needed to address sparsity in AIV forecasting. While human ILI/COVID-19 can be treated as **single-host diffusion on a location graph**, AIV is **multi-host** with **long-range, lineage-linked** introductions along flyways and rapid lineage turnover. A static location graph cannot represent these dependencies. **BLUE** resolves this by coupling a spatial layer with a genetic (case/lineage) layer, which together preserve both spatial and lineage constraints. Modeling both signals simultaneously yields a higher F1 for outbreak detection, whereas simpler models that ignore lineage effects underperform. Please refer to our response to Reviewer zwWG for detailed comparison (https://openreview.net/forum?id=hgj1LQmD09&noteId=YYbMjLodJb).
>
> **2. Justification via Dataset Scale**.
>
> Avian-US covers 3,227 counties, which is larger than standard benchmarks (e.g., Flu-Japan's 47 nodes). High model capacity is required to resolve spatial dependencies at this continental resolution.
>
>
> **3. Generalization beyond avian influenza**.
>
> * **Empirical (ILI).** In **Appendix F.1 (line 1332-1339, page 25)**, we evaluate BLUE on **Flu-Japan** (an ILI dataset). Since it lacks genetics, BLUE **degrades** to a location-centric forecaster and achieves **state-of-the-art performance** (detailed in **Table 7** on page 25), showing generaliability beyond AIV.
> * **Conceptual (variants / other pathogens)**. While our validation centers on AIV, BLUE’s bi-layer heterogeneous topology extends to any pathogen with **strain-/variant-specific dynamics** when *genetic data are available*. Concretely:
> - Location Layer: keep the geographic graph.
> - Case Layer: represent variants (e.g., Delta, Omicron) or strain groups.
> - Genetic edges: weights derived from phylogenetic similarity, enabling the model to differentiate outbreaks driven by highly transmissible or immune-evading variants from those driven by less fit lineages.
>
> This mapping preserves the original mechanism of BLUE and supports downstream forecasting in human diseases (e.g., COVID-19) and other domains (e.g., plant viruses) where variant composition shapes spread.

---

> ### Author Response · Authors · 2025-11-24
> **R2. Clarification of BLUE's Novelty**
>
> We respectfully clarify that the novelty of **BLUE** lies *not in its individual components*, but in their *integrated formulation and theoretical grounding*, which jointly solve a class of forecasting problems **beyond the reach of standard GNN architectures**.
>
> ---
>
> **1. A New Topological Paradigm**.
>
> Prior homogeneous GNNs for epidemiology (e.g., *Cola-GNN* [1], *EpiGNN* [2]) operate on a **static, homogeneous graph of locations**, directly ignoring the *genetic transmission pathway*.
> **BLUE** replaces this paradigm with a bi-layer heterogeneous graph that jointly represents *spatial*, *genetic*, and *ecological* dependencies, providing **a systematic formulation of AIV forecasting as a bi-layer heterogeneous problem** that simultaneously models spatial and genetic transmission.
> The spatial (location) layer captures geography and mobility, and the genetic (case/lineage) layer captures lineage relationships and reassortment-driven connectivity, with cross-layer links encoding introductions between layers.
>
> **2. Principled Structure Learning for Dynamic Graphs**.
>
> Conventional HGNNs (e.g., *HGT* [3]) assume **fixed node sets and relations** during training and inference.
> However, AIV forecasting *violates these assumptions*: **nodes and edges evolve over time** as new cases appear, resolved cases vanish, and lineage relationships update. Message passing tied to a fixed adjacency thus fails to represent the process.
>
> **BLUE** explicitly *decouples* geographic space (Location Layer) from genetic space (Case Layer), enabling the model to track multi-host, long-range pathways that are **invisible** to standard spatio-temporal encodings. A simple concatenation of features or standard heterogeneous GNNs would *discard critical structural information* and fail to handle the dynamic, multi-layer spatio-temporal nature of AIV.
> We introduce an **information-preserving fusion** pipeline that maps the bi-layer graph at each timestep to a unified **Fusion Graph** while retaining lineage-aware dependencies and spatial context. **Ablations (Table 2, page 8)** verify that removing either the genetic edges or the fusion mechanism significantly degrades forecasting accuracy, establishing the necessity of both components in practice.
>
> **3. Theoretical Guarantees for Fusion**.
>
> Our **spectral fusion** method is not a *heuristic*. **Theorem 3.1 (line 317-338)** provides an **upper bound** on the spectral error incurred by fusion, ensuring that the Fusion Graph preserves the *diffusion geometry* of the original multi-layer structure with **linear-in-depth control** of the approximation error.
>
> ---
>
> **BLUE**’s novelty stems from its **topological reformulation**, **dynamic structure learning**, and **theoretical analysis**. They together enable accurate AIV forecasting where standard GNNs fail to capture.
> We hope these points clarify that the contribution is both conceptual and empirical.
> We would also be grateful for any specific citations the reviewer could suggest, to enable direct comparison and to ensure that all relevant prior work is properly acknowledged.
>
> ---
> **Reference**
>
> [1] Cola-GNN: Cross-location attention based graph neural networks for long-term ILI prediction, Proceedings of the 29th ACM international conference on information \& knowledge management, 2020.
>
> [2] EpiGNN: Exploring spatial transmission with graph neural network for regional epidemic forecasting, Joint European Conference on Machine Learning and Knowledge Discovery in Databases, 2022.
>
> [3] Heterogeneous graph transformer, Proceedings of the web conference, 2020.

---

> ### Author Response · Authors · 2025-11-24
> **R3. Eq. (3) explanation**
>
> In EQ. (3), $\mathbf{x}\_{i}^{(spatial)}$ and $\mathbf{x}\_{i}^{(genetic)}$ represent the aggregated spatial and genetic contexts derived from the bi-layer graph. Specifically:
> 1. $\mathbf{x}\_{i}^{(spatial)}$ is the **spatial context vector** for location $i$, computed by applying mean pooling to the feature vectors of all location nodes that are spatially neighboring location $i$.
>
> 2. $\mathbf{x}\_{i}^{(genetic)}$ is the **genetic context vector** for location $i$, computed by applying mean pooling to the feature vectors of all case nodes associated with location $i$.
>
> These two terms represent the aggregated spatial and genetic context derived from the bi-layer graph.
>
> Formally, the fusion node embedding $\mathbf{x}\_{i}$ corresponding to the location node $v\_i^{(c)} \in \mathcal{V}^{(c)}$ at each timestep is generated by
> $$\mathbf{z}^{(c)}\_{i} = f\_{1}\big( \mathbf{x}\_{i}^{(c)} \|\| \mathbf{x}\_{i}^{(spatial)} \big), $$
>
> $$\mathbf{z}^{(p)}\_{i} = f\_{2}\bigl(\mathbf{x}_{i}^{(genetic)}\bigr),$$
>
> $$ \mathbf{x}\_{i} = f\_{m}\bigl(\mathbf{z}^{(c)}\_{i} \|\| \mathbf{z}^{(p)}_{i}\bigr), $$
>
> where$\mathbf{x}\_{i}^{(c)}$ denotes the intrinsic features of location node $i$, and $\|\|$ represents the concatenation operation. The aggregation functions $f\_{1}$, $f\_{2}$, and $f\_{m}$ are implemented as MLPs with nonlinear activation functions.
>
> This architecture integrates diverse node types into coherent fusion embeddings, constructing a semantically rich representation that captures the interface between geographical factors and epidemiological dynamics. This provides a robust foundation for modeling the complex mutual influence of AIV transmission in subsequent modules.
>
> We correct and update Sec. 3.2 in the new version to reflect this precise definition.

---

> ### Author Response · Authors · 2025-11-24
> **R4. Clarification of cross-layer smoothing block**
>
> We thank the reviewer for raising the question of complexity.
> We clarify that while neighbor influence is the mechanism, the end goal of the Cross-Layer Smoothing (CS) block is to **bridge the semantic gap** between the heterogeneous layers prior to fusion.
>
> **1. Mitigating Feature Discrepancy**.
>
> As **described in Section 3.1**, our bi-layer graph couples *two distinct semantic spaces*: the Case Layer (genetic features) and the Location Layer (spatial/ecological features). These features exist in **non-comparable feature spaces**. The CS block employs an MRF-inspired approach to iteratively propagate information across spatial, genetic, and assignment edges. This enforces *local consistency*, ensuring that the representation of a location is semantically aligned with the genetic profile of its associated cases before they are compressed into a single fusion node.
>
>
> **2. Empirical support**.
> This design is necessary for performance. As shown in our **Ablation Study** (**Table 2, page 8**), removing the CS block (*w/o CS*) causes a significant degradation in performance (e.g., F1 drops from 0.1020 to 0.0692). This drop occurs *primarily due to the inherent sparsity and discrepancy of the initial heterogeneous edges* (**line 415-416**). The CS block can regularize these signals and produce coherent representations for the downstream fusion modules.

---

### Official Review · Reviewer_4Tc6 · 2025-11-03

**Soundness:** 1
**Presentation:** 2
**Contribution:** 2
**Rating:** 2
**Confidence:** 4

**Summary:**

This paper introduces BLUE, a bi-layer heterogeneous graph fusion framework for avian influenza forecasting. The proposed model integrates spatial, genetic, and ecological data into a Graph Neural Network (GNN) pipeline that includes the following main components: (1) a bi-layer graph construction that integrates spatial, ecological, and genetic data; (2) a Cross-Layer Smoothing Block (inspired by MRFs) to refine node representations; (3) an Information-Preserving Fusion process that converts the heterogeneous graph into a homogeneous one; and (4) an autoregressive encoder–decoder for multi-step forecasting.

The authors also release a new dataset (Avian-US) combining spatial and genomic features and report performance gains over baselines such as HGT, EpiGNN, and STSGT.

The claimed contributions of this paper are as follows:
1. Propose BLUE, a pipeline that models heterogeneous nodes with multi-type information within a unified framework.
2. Provide a theoretical bound from a spectral perspective for the information-preserving graph fusion, which simplifies heterogeneous graphs without discarding their structural properties.
3. Publicly release the Avian-US dataset and empirically validate BLUE on it, demonstrating superior performance.

**Strengths:**

1. The paper addresses the critical and significant real-world problem of Avian Influenza forecasting, a task with important implications for global biosecurity and public health.

2. A key contribution is the introduction and public release of the Avian-US dataset, a new benchmark for AIV forecasting. This dataset integrates genetic, spatial, and ecological data across thousands of locations, providing a valuable resource for future research.

3. The proposed BLUE pipeline incorporates specific technical components to manage this heterogeneity, such as a cross-layer smoothing block and an information-preserving spectral regularizer.

**Weaknesses:**

1. My main concern is that  the novelty of the proposed model  is limited. The proposed approach, which combines many existing techniques into one large system, feels more like a complex engineering integration. It lacks the simplicity and fundamental novelty typically valued by the ICLR community.

2. The proposed model is overly complicated, with insufficient ablation to support each component. The BLUE pipeline combines many existing techniques (R-GCN-like smoothing, LSH sampling, attention gating, spectral loss, autoregressive decoder). The ablation study, while decent, does not disentangle these components.

3. The validation on Flu-Japan data is meaningless. The second experiment, on Flu-Japan, is invalid. By simulating a uniform, homogeneous “case layer,” the authors test a model component that is completely different from what they propose. This experiment fails to validate the paper’s core thesis about fusing genetic data and should be removed or completely reframed. As the authors mention: “We first construct the location layer based on the provided adjacency matrix. Each location node is assigned features based on the reported infection counts across prefectures. To represent case-level information, we simulate case nodes and associate them with their respective infected locations.”

Given this setup, the evaluation on Flu-Japan does not test real heterogeneous data. Thus, the evaluation relies on a single synthetic dataset that does not reflect the claimed capabilities of BLUE.

a. What is the scientific or empirical value of testing a heterogeneous fusion model on a homogeneous, simulated dataset?
b. How does BLUE generalize to other human epidemiological datasets (e.g., influenza-like illness or COVID-19 variants)?
c. How is “genetic similarity” translated into edge weights—through a fixed threshold, adaptive kernel, or another method?


4. The fusion process is unnecessarily complex (LSH sampling + attention gating + spectral regularization). Could the authors provide an ablation that isolates the contribution of the LSH and attention components? For example, what is the performance if one uses a simple k-NN graph on the fused node embeddings but keeps the spectral regularizer? This would clarify whether the complex edge-generation mechanism actually contributes to performance improvements.

**Questions:**

see weaknesses

---

> ### Author Response · Authors · 2025-11-24
> **R1. Clarification on Novelty and Contributions of BLUE**
>
> We respectfully clarify that the novelty of **BLUE** lies *not in its individual components*, but in their *integrated formulation and theoretical grounding*, which jointly solve a class of forecasting problems **beyond the reach of standard GNN architectures**.
>
> ---
>
> **1. A New Topological Paradigm**.
>
> Prior homogeneous GNNs for epidemiology (e.g., *Cola-GNN* [1], *EpiGNN* [2]) operate on a **static, homogeneous graph of locations**, directly ignoring the *genetic transmission pathway*.
> **BLUE** replaces this paradigm with a bi-layer heterogeneous graph that jointly represents *spatial*, *genetic*, and *ecological* dependencies, providing **a systematic formulation of AIV forecasting as a bi-layer heterogeneous problem** that simultaneously models spatial and genetic transmission.
> The spatial (location) layer captures geography and mobility, and the genetic (case/lineage) layer captures lineage relationships and reassortment-driven connectivity, with cross-layer links encoding introductions between layers.
>
> **2. Principled Structure Learning for Dynamic Graphs**.
>
> Conventional HGNNs (e.g., *HGT* [3]) assume **fixed node sets and relations** during training and inference.
> However, AIV forecasting *violates these assumptions*: **nodes and edges evolve over time** as new cases appear, resolved cases vanish, and lineage relationships update. Message passing tied to a fixed adjacency thus fails to represent the process.
>
> **BLUE** explicitly *decouples* geographic space (Location Layer) from genetic space (Case Layer), enabling the model to track multi-host, long-range pathways that are **invisible** to standard spatio-temporal encodings. A simple concatenation of features or standard heterogeneous GNNs would *discard critical structural information* and fail to handle the dynamic, multi-layer spatio-temporal nature of AIV.
> We introduce an **information-preserving fusion** pipeline that maps the bi-layer graph at each timestep to a unified **Fusion Graph** while retaining lineage-aware dependencies and spatial context. **Ablations (Table 2, page 8)** verify that removing either the genetic edges or the fusion mechanism significantly degrades forecasting accuracy, establishing the necessity of both components in practice.
>
> **3. Theoretical Guarantees for Fusion**.
>
> Our **spectral fusion** method is not a *heuristic*. **Theorem 3.1 (line 317-338)** provides an **upper bound** on the spectral error incurred by fusion, ensuring that the Fusion Graph preserves the *diffusion geometry* of the original multi-layer structure with **linear-in-depth control** of the approximation error.
>
> ---
>
> **BLUE**’s novelty stems from its **topological reformulation**, **dynamic structure learning**, and **theoretical analysis**. They together enable accurate AIV forecasting where standard GNNs fail to capture.
> We hope these points clarify that the contribution is both conceptual and empirical.
> We would also be grateful for any specific citations the reviewer could suggest, to enable direct comparison and to ensure that all relevant prior work is properly acknowledged.
>
> ---
> **Reference**
>
> [1] Cola-GNN: Cross-location attention based graph neural networks for long-term ILI prediction, Proceedings of the 29th ACM international conference on information \& knowledge management, 2020.
>
> [2] EpiGNN: Exploring spatial transmission with graph neural network for regional epidemic forecasting, Joint European Conference on Machine Learning and Knowledge Discovery in Databases, 2022.

---

> ### Author Response · Authors · 2025-11-24
> **R2. Complexity and further ablation study**
>
> Thank you for raising the question of complexity. For a multi-scale system like AIV, additional machinery is only problematic if **redundant**. To demonstrate that each component of **BLUE** is **functional** rather than decorative, we extend our ablation analysis as follows.
>
> **1. Correction on Table 2**:
> We identified and fixed a labeling oversight in the original **Table 2 (p. 8)**: *w/o CS+gen* should read *w/o CS+Spec*, consistent with removing both Cross-Layer Smoothing and the Spectral Regularizer and *only implementing the auto-regressive encoder-decoder framework*, as stated in **Sec. 4.3** (**lines 405–406**). The original ablations therefore already validate the roles of **Cross-Layer Smoothing** (*w/o CS*), the **Spectral Regularizer** (w/o Spec), and **autoregressive forecasting** (*w/o CS+Spec*).
>
> **2. Additional ablations isolating sampler vs. gate**:
> We further introduce two controlled variants (all other components unchanged):
> - **Is the LSH sampler just for speed?** (Tested via *w/ attn* - Full Dense Attention).
> - **Is the Attention Gate necessary?** (Tested via *w/ LSH* - LSH with simple averaging).
>
> **Findings:**
>
> 1. **LSH is better than all-pairs**: Moving to dense attention (*w/ attn*) reduces F1 (from 0.1001 to 0.0901), indicating that LSH acts as a **structural regularizer**: it restricts aggregation to high-likelihood neighborhoods and avoids overfitting to noisy, long-range correlations common in sparse epidemiological settings.
>
> 2. **The attention gate filters noise and reweights signals**: Removing the gate (*w/ LSH*) sharply increases MAE (from 0.0848 to 0.1007). This confirms that Attention Gate is crucial for weighing spatial vs. genetic influence dynamically.
>
>
> The full BLUE consistently outperforms simplified variants. LSH first concentrates candidates into strong neighborhoods. The gate then adaptively aggregates them. The combination yields the most consistent gains in all metrics, justifying the designs.
>
>
> | Model        | w/ LSH          | w/ attn         | **BLUE**            |
> | ------------ | --------------- | --------------- | ------------------- |
> | **RMSE (↓)** | 0.6141 ± 0.0630 | 0.6157 ± 0.0824 | **0.6106 ± 0.0719** |
> | **MAE (↓)**  | 0.1007 ± 0.0348 | 0.0945 ± 0.0471 | **0.0848 ± 0.0182** |
> | **PCC (↑)**  | 0.0855 ± 0.0109 | 0.0755 ± 0.0112 | **0.0871 ± 0.0132** |
> | **SCC (↑)**  | 0.1105 ± 0.0110 | 0.1114 ± 0.0163 | **0.1217 ± 0.0140** |
> | **F1 (↑)**   | 0.0994 ± 0.0264 | 0.0901 ± 0.0308 | **0.1001 ± 0.0193** |
>
> **Table 1.** Additional ablation results

---

> ### Author Response · Authors · 2025-11-24
> **R3. a. Empirical Validation on Flu-Japan dataset; b. generalizability of BLUE to human epidemiological datasets; c. the calculation of edge weights**
>
> We appreciate the reviewer’s critique and wish to clarify the specific scope and purpose of the Flu-Japan experiment. Specifically, experiments on Flu-Japan are designed to test **architectural robustness**, not genetic fusion. It answers a critical question for real-world deployment: *Does BLUE fail when genomic data is unavailable or uninformative*?
>
> **(a) Scientific value of testing on Flu-Japan**
>
> - **Flu-Japan is not synthetic.** It is a widely used homogeneous benchmark for ILI forecasting. We include it as a *generalization and robustness check* complementary to our heterogeneous **Avian-US** setting.
>
> - **Information parity across methods**. Because Flu-Japan *lacks case-level genetics*, we omit the case layer solely to ensure equivalent information content across BLUE and baselines:
>
>    - **Baselines**: we add a constant “genetic similarity” feature to location nodes.
>    - **BLUE**: We connect case nodes with uniform genetic weights, so the case layer contains no extra information.
>
>    Under this setup, both sides receive the same effective information, and the test isolates whether BLUE degrades gracefully when genetic structure is uninformative.
>
> - **Comparable performance on a standard benchmark**. As reported in **Table 7** (**p. 25, Appendix F**), BLUE matches or surpasses strong ILI baselines (e.g., Cola-GNN, Epi-Cola-GNN), demonstrating competitiveness on human disease data while adding the capability to exploit genetics when available.
>
> **(b) Generalization to human epidemiology**
>
> - **Empirical evidence**. Flu-Japan provides direct evidence that BLUE generalizes to a human ILI dataset and performs at state-of-the-art levels (shown in **Table 7**).
>
> - **Conceptual fit for variant-aware diseases**. The bi-layer design can transfer naturally to human disease spread settings (e.g., COVID-19):
>    - Location layer: unchanged.
>    - Case layer: nodes represent cases and/or variants (e.g., Delta, Omicron).
>    - Genetic edges: encode variant similarity among variants, enabling BLUE to model how different variants impact transmission risk.
> This extends standard location-only models by capturing variant-specific dynamics.
>
> **(c) Construction of genetic similarity and edge weights**
> We do not use a fixed threshold or adaptive kernel. As detailed in **Sec. 3 (lines 190–197) and App. D.3**, edge weights are a direct transformation of evolutionary distance derived from the Kimura 2-parameter (K80) model.
>    - We calculate the evolutionary distance $d$ between sequences based on transition/transversion rates.
>    - The edge weight is computed as $w_{kl}^{(ge)} = 1 - d$, normalized to $[0,1]$), where higher $w_{kl}^{(ge)}$ indicate lower transition/transversion rates and closer genetic ancestry.
>
> We agree that the Flu-Japan dataset does not evaluate genetic fusion directly. Instead, it serves to assess model robustness and parity with baselines in a homogeneous setting. We retain this experiment because it demonstrates (i) graceful degradation when genetic information is absent or uninformative, and (ii) competitive accuracy on a widely used benchmark for human influenza forecasting. We will clarify the experiments in the revised version.

---

> ### Author Response · Authors · 2025-11-24
> **R4. Ablation comparison of KNN**
>
> We appreciate the reviewer’s suggestion to isolate the edge-generation components. To address this, we implemented the requested ablation, replacing the LSH+Attention block with a **k-NN graph constructor (k=20)** while retaining the **Spectral Regularizer**. We argue that the complexity of the BLUE fusion module is justified by two critical factors: *computational scalability* and *empirical performance*.
>
> **1. Scalability**:
> The LSH sampler is not merely a heuristic. Instead,  it serves as a complexity reduction mechanism. Constructing a standard k-NN graph requires all-pairs distance computations, leading to $O(N^2)$ complexity at every timestep. For our dataset ($N=3,227$), this is computationally expensive for real-time training. LSH approximates this neighborhood search in $O(N \log N)$ (detailed in **Appendix A.3.2, page 14-15**).
>
> **2. Ablation performance**:
> We add three variants: w/ attn (Full Dense Attention), w/ LSH (LSH with simple averaging), and *w/ KNN* (replaces LSH + attention with a k-NN graph on fused embeddings with $k=20$. We keep the spectral regularizer and all other components unchanged.
>
> **Findings**:
> - The *w/ attn* variant performs poorly on trend metrics (SCC/PCC), indicating that without a sampler, the model is **overwhelmed by correlated noise** from long-range connections. LSH is crucial for restricting the model's focus to **a manageable, relevant neighborhood.**
>
> - While k-NN captures ranking trends (SCC) reasonably well, it fails to predict infection counts accurately, as evidenced by its high MAE (0.1057 vs. BLUE's 0.0848). This is likely because k-NN forces *a fixed number of connections* even for isolated nodes (**over-linking**), mixing heterogeneous signals.
>
> - BLUE avoids this by using **LSH to find candidates** and **Attention Gate to down-weight irrelevant ones**. As shown in Table. 1, BLUE outperforms all variants. It achieves the best performance, confirming that the combination of efficient sampling and learnable gating is essential for accurate outbreak detection.
>
> Table. 1. Additional ablation results.
>
> | Model        | w/ LSH          | w/ attn         | w/ KNN          | **BLUE**            |
> | ------------ | --------------- | --------------- | --------------- | ------------------- |
> | **RMSE (↓)** | 0.6141 ± 0.0630 | 0.6157 ± 0.0824 | 0.6146 ± 0.0701 | **0.6106 ± 0.0719** |
> | **MAE (↓)**  | 0.1007 ± 0.0348 | 0.0945 ± 0.0471 | 0.1057 ± 0.0238 | **0.0848 ± 0.0182** |
> | **PCC (↑)**  | 0.0855 ± 0.0109 | 0.0755 ± 0.0112 | 0.0847 ± 0.0139 | **0.0871 ± 0.0132** |
> | **SCC (↑)**  | 0.1105 ± 0.0110 | 0.1114 ± 0.0163 | 0.1201 ± 0.0119 | **0.1217 ± 0.0140** |
> | **F1 (↑)**   | 0.0994 ± 0.0264 | 0.0901 ± 0.0308 | 0.0963 ± 0.0201 | **0.1001 ± 0.0193** |

---

### Official Review · Reviewer_zwWG · 2025-11-11

**Soundness:** 2
**Presentation:** 2
**Contribution:** 2
**Rating:** 2
**Confidence:** 4

**Summary:**

This paper formulates avian influenza forecasting problem by proposing a Bi-Layer heterogeneous graph fUsion pipEline (BLUE). This pipeline integrates genetic, spatial, and ecological data to achieve accurate outbreak forecasting. The Avian-US dataset has been released, and BLUE achieves superior performance over existing baselines.

**Strengths:**

1. This paper deals with a very important problem in the real world.

2. The authors release a new dataset for benchmarking avian influenza forecasting. This contribution should be very meaningful for the advancement of this field.

3. The proposed framework, BLUE, shows good engineering with commonly-used techniques.

**Weaknesses:**

1. It is not very clear why the avian influenza forecasting needs more specialized methods while the conventional influenza-like and COVID-19 forecasting is not.

2. The proposed framework is a combination of existing techniques, which may not have sufficient novelty. As far as I understand, the novelty is importantly regarded in ICLR. This manuscript may not perfectly suit this venue.

3. The proposed framework seems to be overly complicated, in comparison to the size of the dataset that will be used for the framework. The framework’s complexity and heavy computation may also limit its scalability and real-time applicability in operational surveillance systems.

4. The practicability of the proposed framework is questionable, because it would be very hard to obtain the genetic information needed to build the case layer.

**Questions:**

See the weaknesses.

---

> ### Author Response · Authors · 2025-11-23
> **R1. Differences between AIV and human-like diseases.**
>
> We appreciate the reviewer's valuable concern.
> Conventional ILI and COVID-19 forecasters assume diffusion on **a single spatial layer** driven by geographic proximity or human mobility [1–3]. These assumptions work for human-centric diseases but break for AIV, where spread is **multi-host** and often **seeded by migratory birds** that link distant regions along flyways [5–7]. **BLUE** resolves this mismatch with a bi-layer design.
>
> 1. **Single-Host vs. Multi-Host Transmission Dynamics**
>
>    Standard ILI/COVID-19 forecasting treats spread as a **single-host** process evolving on a **single-layer** location graph, where risk diffuses between geographically adjacent regions or along human mobility networks (e.g., commuting flows) [1–3]. In that setting, spatial proximity and mobility volumes serve as proxies for transmission risk.
>
>    By contrast, AIV is **wildlife-driven** and **multi-host** [4]. Outbreaks are frequently seeded by migratory wild birds moving along continental flyways, which induce **long-range transmission edges** between distant regions that are *not captured by geographic adjacency* [5]. Frequent introductions across the *wildlife–poultry–environment* interface further decouple risk from human mobility patterns [5–7]. A single-layer, location-only graph therefore misrepresents these dependencies and can incorrectly treat distant, genetically related outbreaks as independent. BLUE addresses this by **adding genetic edges** that encode non-spatial introductions inferred from lineage relationships.
>
> 2. **Latent Genetic Dependencies**.
>
>    Unlike human ILI forecasting, where case-level viral evolution is often abstracted into aggregate counts, AIV dynamics are driven by *high-frequency reassortment and lineage turnover* [6, 7].
>    Two geographically distant cases may be **epidemiologically identical** if they share a specific viral lineage, while two neighboring cases may represent distinct, unrelated outbreaks.
>    We refer to these lineage-based connections as **invisible linkages**: dependencies that live in *genetic space* but are not observable in *geographic space*.
>    Models that treat all cases as homogeneous counts (without lineage structure) discard this signal and can misestimate both risk and timing [3,8]. **BLUE** makes these dependencies explicit via a genetic layer and cross-layer information flow.
>
> 3. **Empirical Validation in Sec. 4**.
>    Our ablations (Table 2) show that spatial proximity alone is insufficient for AIV forecasting. Removing the **genetic edges** (*w/o gen*) or disabling **cross-layer smoothing** (*w/o CS*) leads to a statistically significant performance decline. This aligns with prior evidence that combining spatial dynamics with phylogeographic linkages improves inference and prediction in influenza systems [9]. Collectively, these results support the necessity of BLUE’s bi-layer design to capture heterogeneous AIV transmission pathways.
>
> ---
>
> **References**
>
> [1] Forecasting the spatial transmission of influenza in the United States, PNAS 2018.
>
> [2] Enhancing spatial spread prediction of infectious diseases through integrating multi-scale human mobility dynamics, Proceedings of the 31st ACM International Conference on Advances in Geographic Information Systems, 2023.
>
> [3] Forecasting epidemic spread with recurrent graph gate fusion transformers, IEEE Journal of Biomedical and Health Informatics, 2024.
>
> [4] Forecasting influenza activity using machine-learned mobility map, Nature Communications, 2021.
>
> [5] Transmission dynamics of highly pathogenic avian influenza virus at the wildlife-poultry-environmental interface: a case study, One Health, 2024.
>
> [6] Rapid evolution of A (H5N1) influenza viruses after intercontinental spread to North America, Nature Communications, 2023.
>
> [7] Transatlantic spread of highly pathogenic avian influenza H5N1 by wild birds from Europe to North America in 2021, Scientific reports, 2022.
>
> [8] Cola-GNN: Cross-location attention based graph neural networks for long-term ILI prediction, Proceedings of the 29th ACM international conference on information \& knowledge management, 2021.
>
> [9] Integrating dynamical modeling and phylogeographic inference to characterize global influenza circulation, PNAS nexus, 2025.

---

> ### Author Response · Authors · 2025-11-24
> **R2. Clarification on Novelty and Contributions of BLUE**
>
> We respectfully clarify that the novelty of **BLUE** lies *not in its individual components*, but in their *integrated formulation and theoretical grounding*, which jointly solve a class of forecasting problems **beyond the reach of standard GNN architectures**.
>
> ---
>
> **1. A New Topological Paradigm**.
>
> Prior homogeneous GNNs for epidemiology (e.g., *Cola-GNN* [1], *EpiGNN* [2]) operate on a **static, homogeneous graph of locations**, directly ignoring the *genetic transmission pathway*.
> **BLUE** replaces this paradigm with a bi-layer heterogeneous graph that jointly represents *spatial*, *genetic*, and *ecological* dependencies, providing **a systematic formulation of AIV forecasting as a bi-layer heterogeneous problem** that simultaneously models spatial and genetic transmission.
> The spatial (location) layer captures geography and mobility, and the genetic (case/lineage) layer captures lineage relationships and reassortment-driven connectivity, with cross-layer links encoding introductions between layers.
>
> **2. Principled Structure Learning for Dynamic Graphs**.
>
> Conventional HGNNs (e.g., *HGT* [3]) assume **fixed node sets and relations** during training and inference.
> However, AIV forecasting *violates these assumptions*: **nodes and edges evolve over time** as new cases appear, resolved cases vanish, and lineage relationships update. Message passing tied to a fixed adjacency thus fails to represent the process.
>
> **BLUE** explicitly *decouples* geographic space (Location Layer) from genetic space (Case Layer), enabling the model to track multi-host, long-range pathways that are **invisible** to standard spatio-temporal encodings. A simple concatenation of features or standard heterogeneous GNNs would *discard critical structural information* and fail to handle the dynamic, multi-layer spatio-temporal nature of AIV.
> We introduce an **information-preserving fusion** pipeline that maps the bi-layer graph at each timestep to a unified **Fusion Graph** while retaining lineage-aware dependencies and spatial context. **Ablations (Table 2, page 8)** verify that removing either the genetic edges or the fusion mechanism significantly degrades forecasting accuracy, establishing the necessity of both components in practice.
>
> **3. Theoretical Guarantees for Fusion**.
>
> Our **spectral fusion** method is not a *heuristic*. **Theorem 3.1 (line 317-338)** provides an **upper bound** on the spectral error incurred by fusion, ensuring that the Fusion Graph preserves the *diffusion geometry* of the original multi-layer structure with **linear-in-depth control** of the approximation error.
>
> ---
>
> **BLUE**’s novelty stems from its **topological reformulation**, **dynamic structure learning**, and **theoretical analysis**. They together enable accurate AIV forecasting where standard GNNs fail to capture.
> We hope these points clarify that the contribution is both conceptual and empirical.
> We would also be grateful for any specific citations the reviewer could suggest, to enable direct comparison and to ensure that all relevant prior work is properly acknowledged.
>
> ---
> **Reference**
>
> [1] Cola-GNN: Cross-location attention based graph neural networks for long-term ILI prediction, Proceedings of the 29th ACM international conference on information \& knowledge management, 2020.
>
> [2] EpiGNN: Exploring spatial transmission with graph neural network for regional epidemic forecasting, Joint European Conference on Machine Learning and Knowledge Discovery in Databases, 2022.
>
> [3] Heterogeneous graph transformer, Proceedings of the web conference, 2020.

---

> ### Author Response · Authors · 2025-11-24
> **R3. Complexity, Scalability, and Feasibility of BLUE**
>
> We thank the reviewer for this opportunity to clarify the design rationale regarding model complexity, scalability, and feasibility.
>
> **1. Dataset Scale (3,227 vs. 47 Nodes)**.
>
> The dataset is large-scale and granular. **Avian-US** covers **N=3,227 U.S. counties**. This is significantly larger than standard benchmarks like **Flu-Japan** (**47 nodes**) or **US-States** (**49 nodes**) [1]. Moreover, the graph is *dynamic*, incorporating a time-varying set of individual case nodes at every timestep. This large-scale and dynamic structure necessitates a comprehensive pipeline capable of fusing complex, multi-source information.
>
> **2. Complexity is Managed, Not Unbounded**.
>
> We explicitly designed **BLUE** to avoid the $O(N^2)$ bottleneck of standard graph construction. As detailed in **Appendix A.3** (**line 731-774, page 14-15**), we employ an LSH-based sampler that restricts comparisons to likely neighbors, reducing complexity *from quadratic to $O(N \log N)$*. Furthermore, as noted in **Appendix A.4** (line776-808, page 15), the computationally intensive autoregressive forecasting operates on the compressed fusion graph (size $N$), making the forecasting complexity **independent of the case-node count** $M_t$. This ensures the model remains scalable even as the number of genomic samples grows.
>
> **3. Complexity is Justified by Biology**.
>
> The bi-layer structure is not *"over-engineering"*: it is the **minimal topological formulation** required to capture AIV transmission. AIV transmission is driven by two distinct forces: *spatial proximity* and *genetic lineage*. Standard homogeneous GNNs cannot represent these two distinct pathways simultaneously.
> Our ablation study (**Table 2, page 8**) proves that simplifying the model (removing genetic links) destroys predictive performance, confirming that the current architecture is necessary for accuracy.
>
> **4. Operational Feasibility**.
> In the context of epidemiological surveillance, 'real-time' is defined by the timeliness required for effective public health intervention, typically operating on daily or weekly decision cycles rather than the millisecond latency standards of computational engineering [2]. As detailed in **Appendix E.2 (Page 22)**, **BLUE** is successfully trained and evaluated using standard hardware (e.g., a single NVIDIA V100, DGX A100, or RTX A5000) within hours. This demonstrates that the model fits well within the computational budget for routine daily or weekly retraining on standard GPU infrastructure.
>
> **Reference**
>
> [1] Cola-GNN: Cross-location attention based graph neural networks for long-term ILI prediction, Proceedings of the 29th ACM international conference on information \& knowledge management, 2020.
>
> [2] Using digital surveillance tools for near real-time mapping of the risk of infectious disease spread, npj Digital Medicine, 2021.

---

> ### Author Response · Authors · 2025-11-24
> **R4. Practicability of genetic information**
>
> We clarify that the integration of genetic information is highly practicable and relies entirely on publicly available, routinely collected data. Genomic sequencing has become a standard protocol for global AIV surveillance (unlike seasonal influenza, where it is less ubiquitous). As detailed in **Appendix D.2 and D.3** (line 1026-1088, page 20-21) and our reference, the genomic data used in **BLUE** is sourced entirely from the **NCBI GenBank sequence database** (link: https://www.ncbi.nlm.nih.gov/genbank/). Labs worldwide routinely upload genomic data to the database as part of standard public health monitoring, making this data readily accessible for operational systems without additional collection cost.

---

### Author Response · Authors · 2025-11-30
**Rebuttal Summary**

Dear Area Chair,

We are writing to respectfully summarize the rebuttal and discussion period for our submission. We thank the reviewers zwWG, 4Tc6, and kX83 for their constructive feedback. During the rebuttal, we have **extended our ablation studies**, **clarified theoretical contributions**, and **corrected factual misunderstandings**.
We believe the questions have been reasonably resolved and the manuscript has been significantly strengthened.

---

Our rebuttal directly addressed all major points:

**1. Clarification on Novelty** (Response to **Reviewer zwWG’s Q1\& Q2, Reviewer 4Tc6’s Q1, Reviewer kX83’s Q2 \& Q4**): we clarified that BLUE is not a combination of existing techniques, but a systematic solution to the unique challenges of AIV forecasting.
   - **Distinct Multi-Host Dynamics:** Existing SOTA models (for ILI/COVID) assume "single-host" diffusion driven by geographic proximity or human mobility. In contrast, AIV is a **multi-host system** driven by migratory birds, creating "invisible linkages" (lineage-based connections) across distant regions. BLUE resolves the fundamental failure of spatial-only models to capture these non-spatial, long-range dependencies.
   - **A Topological Paradigm Shift:** Unlike traditional epidemiological GNNs (e.g., Cola-GNN) that operate on **static, homogeneous graphs**, BLUE introduces a **dynamic, bi-layer heterogeneous paradigm,** effectively modelling the **evolving nodes and edges** inherent to AIV—a dynamic structure that static graph models cannot represent.
   - **Theoretical Analysis:**  We further emphasized our **theoretical analysis (Theorem 3.1)**, which provides **upper bound on the spectral error** incurred by fusion, ensuring it is a mathematically grounded solution rather than a heuristic combination.
   - **Bridging the Semantic Gap**: We clarified that the Cross-Layer Smoothing block addresses the unique challenge of fusing **non-comparable feature spaces** (genetic sequences vs. spatial distances), with ablations confirming its necessity (Table. 2).

**2. Justification of Complexity via New Ablations** (Response to **Reviewer zwWG’s Q3, Reviewer 4Tc6’s Q2 \& Q4, Reviewer kX83’s Q1**):
   - **Empirical Necessity:** The results (Table 1 in rebuttal) show that k-NN significantly underperforms BLUE due to "over-linking" noise. In contrast, BLUE’s LSH sampler acts as a **structural regularizer**, effectively filtering long-range noise.
   - **Scalability:** We clarified that LSH is a complexity reduction mechanism ($O(N \log N)$), enabling scalability on our large-scale graph ($N=3,227$), whereas standard k-NN scales quadratically ($O(N^2)$).
   - **Complexity-Performance Trade-off:** We provided a formal complexity analysis showing that BLUE scales **near-linearly ($O(N \log N)$)** due to LSH sampling. In contrast, baselines like Cola-GNN scale **quadratically ($O(N^2)$) or cubically** ($O(N^3)$).

**3. Resolution of Validity & Generalization Concerns** (Response to **Reviewer 4Tc6’s Q3)**

- **Flu-Japan Experiment:** We clarified a misunderstanding regarding the Flu-Japan benchmark. It is intended as a **robustness check,** demonstrating the reasonable degradation when genetic data is absent, rather than a test for genetic fusion.
- **Applicability:** We highlighted that BLUE’s bi-layer design is conceptually transferable to other variant-aware diseases (e.g., **COVID-19**), where the "Case Layer" maps to viral variants (e.g., Delta/Omicron).

**4. Clarification of Misunderstandings** (Response to **Reviewer zwWG’s Q4, Reviewer kX83’s Q1):**

- **Data Availability:** We corrected the statement that genetic data is hard to obtain. We rely on standard **public repositories (NCBI/GenBank)** and provide a reproducible record linkage pipeline (Appendix D.2).
- **Dataset Scale:** We pointed out that our dataset covers **3,227 U.S. counties**, significantly larger than standard benchmarks (e.g., Flu-Japan's 47 nodes), necessitating the efficient architecture we proposed.

In conclusion, our work addresses **a critical gap in multi-host epidemic forecasting** through **BLUE**, a bi-layer heterogeneous framework that uniquely integrates spatial and genetic dynamics with theoretical guarantees. During the rebuttal, we resolved concerns regarding novelty and complexity, and substantially strengthened our empirical evaluation by adding new ablation studies. We have **revised the manuscript to incorporate these clarifications and additional results**.

Thanks very much for your time and effort dedicated to reviewing our work and coordinating the discussion process.

---

### Note · Authors · 2025-12-01

**Comment:**

Dear Area Chair and Reviewers,

After careful consideration of the reviews and the current status of the discussion, we have decided to withdraw our submission from ICLR 2025.

We sincerely thank the reviewers for their time and constructive feedback. We particularly appreciate that the reviewers acknowledged the significant difficulty and importance of the multi-host Avian Influenza forecasting problem, as well as the novelty and theoretical contribution of our proposed bi-layer topological paradigm.

While we believe the extensive new experiments and clarifications provided during the rebuttal have strengthened the paper, we recognize that the current scores remain low. With the discussion phase coming to a close and reviewers can no longer engage, we decide to withdraw the paper to allow us the time to fully incorporate this valuable feedback and refine the manuscript for submission to another venue.

Thank you again for helping us improve our research.

Sincerely,

Authors of Submission 11513

**Withdrawal Confirmation:**

I have read and agree with the venue's withdrawal policy on behalf of myself and my co-authors.